# FedSMU: Communication-Efficient and Generalization-Enhanced Federated Learning through Symbolic Model Updates

**Xinyi Lu** [* 1]   **Hao Zhang** [* 1]   **Chenglin Li** [1]   **Weijia Lu** [2]   **Zhifei Yang** [2]   **Wenrui Dai** [1]   **Xiaodong Zhang** [2]   **Xiaofeng Ma** [2]   **Can Zhang** [2]   **Junni Zou** [1]   **Hongkai Xiong** [1]

## Abstract

The significant communication overhead and client data heterogeneity have posed an important challenge to current federated learning (FL) paradigm. Existing compression-based and optimization-based FL algorithms typically focus on addressing either the model compression challenge or the data heterogeneity issue individually, rather than tackling both of them. In this paper, we observe that by symbolizing the client model updates to be uploaded (i.e., normalizing the magnitude for each model parameter at local clients), the model heterogeneity, essentially stemmed from data heterogeneity, can be mitigated, and thereby helping improve the overall generalization performance of the globally aggregated model at the server. Inspired with this observation, and further motivated by the success of Lion optimizer in achieving the optimal performance on most tasks in the centralized learning, we propose a new FL algorithm, called FedSMU, which simultaneously reduces the communication overhead and alleviates the data heterogeneity issue. Specifically, FedSMU splits the standard Lion optimizer into the local updates and global execution, where only the symbol of client model updates commutes between the client and server. We theoretically prove the convergence of FedSMU for the general non-convex settings. Through extensive experimental evaluations on several benchmark datasets, we demonstrate that our FedSMU algorithm not only reduces the communication overhead, but also achieves a better generalization performance than the other compression-based and optimization-based baselines.

---

[*]Equal contribution  [1]Shanghai Jiao Tong University, Shanghai, China. [2]United Automotive Electronic Systems, Shanghai, China. Correspondence to: Chenglin Li, Wenrui Dai, and Junni Zou <lcl1985@sjtu.edu.cn>.

*Proceedings of the $42^{nd}$ International Conference on Machine Learning*, Vancouver, Canada. PMLR 267, 2025. Copyright 2025 by the author(s).

## 1. Introduction

Federated learning (FL) is a large-scale machine learning paradigm wherein a multitude of clients, under the orchestration of a central server, collaboratively learn a model without the need of sharing or exchanging any raw client data (McMahan et al., 2017). This paradigm is commonly adopted in data-constrained or data-sensitive environments, such as Internet of things (IoT), healthcare, and finance (Khan et al., 2021; Rieke et al., 2020; Yang et al., 2019; Haibo et al., 2023). In essence, FL is distinguished from the traditional distributed learning in the following three major challenges. **High communication cost.** During each communication round of training, the clients are required to transmit their local model parameters (or updates) to the central server for global aggregation. When the number of model parameters becomes significantly large, this transmission process may result in a huge bandwidth consumption. **Data heterogeneity.** Due to the inherently private and personalized nature of federated clients, the datasets across these clients tend to exhibit distinct statistical distributions. Such a data heterogeneity may introduce significant biases into the globally aggregated model, consequently impairing its generalization performance. **Partial client participation.** In practical scenarios, clients may join or leave the FL system at random time intervals. This highly dynamic behavior results in only a small subset of clients being active for training during each communication round.

To address these challenges, extensive exploration has been conducted in the FL community, but from different perspectives. On one hand, compression-based federated algorithms aim to reduce the amount of data required for model parameter (or update) transmission. For instance, quantization compression, such as signSGD (Bernstein et al., 2018a;b), QSGD (Alistarh et al., 2017) and FedPAQ (Reisizadeh et al., 2020), quantize the gradient values into lower-precision integers, thereby reducing the number of transmitted bits. While the sparsity compression methods typically sparsify the gradient vector by setting some of its elements to zero or fewer bits, with the aim of reducing the data transmission cost (Wangni et al., 2018; Aji & Heafield, 2017; Lin et al., 2017). However, a direct application of compression methods may

lead to information loss, resulting in decreased model accuracy (Yu et al., 2022) and slower convergence rates, or even divergence of the training process (Beznosikov et al., 2023). To mitigate these issues, strategies such as error feedback (Richtárik et al., 2021) have been developed, which incorporate residual errors from previous compression steps into the optimization process. While many algorithms with the error feedback require the full participation of the clients, and if only partial clients are involved, their performance will degrade (Li & Li, 2023).

On the other hand, several optimization-based federated algorithms, have been proposed to address the data heterogeneity issue. For example, SCAFFOLD (Karimireddy et al., 2020) aims to mitigate the client variance by designing and iteratively updating the control variates. Though theoretically effective, it incurs doubling the communication overhead. FedGen (Venkateswaran et al., 2023) regulates the local training by transmitting additional generators. Several adaptive algorithms (Tong et al., 2020) dynamically adjust their learning rates based on the divergence between the local and global models, thereby enhancing the generalization performance in federated settings. Most of these optimization-based FL algorithms, which mainly aim at mitigating the data heterogeneity, may incur additional communication overhead of information exchange for further performance improvement.

In this paper, we aim to design an algorithm capable of simultaneously addressing the communication bottleneck and data heterogeneity, without being constrained by the partial client participation issue. To achieve this goal, we first revisit the fundamental FedAvg algorithm and identify that heterogeneous magnitudes of model updates may result in certain clients' updates being overlooked, thus leading to an unstable and sub-optimal aggregation of the global model. Building upon this observation, we then introduce the concept of Magnitude Uniformity (MU) index, which quantifies the clients' contribution to the global model's update. We empirically validate that this MU index is influenced by the degree of data heterogeneity in FL, indicating that a more heterogeneous data distribution causes a greater heterogeneity in the magnitudes of client model updates. Furthermore, heterogeneous client updates may contribute to a decline in the global model's generalization performance. To address this issue and further reduce communication overhead, we are motivated to symbolize the model updates as an immediate solution, and propose the FedSMU algorithm. Our contributions can be summarized as follows.

- We develop a compression-based FL method, FedSMU. It uses the sign operation to achieve 1-bit compression and thus greatly saves the communication cost. Simultaneously, we leverage the design of Lion optimizer (Chen et al., 2024) to enhance the generalization performance while maintaining the benefits of compression.

- We conduct a convergence analysis of FedSMU under the general non-convex settings, and find its convergence rate as $\mathcal{O}(\frac{1}{\sqrt{T}})$, where $T$ is the total number of communication rounds. This theoretical result matches with the convergence rates of existing FL algorithms.

- We conduct a series of experiments to demonstrate the superiority of FedSMU. By comparing FedSMU with the other compression-based and optimization-based FL algorithms, we show that our FedSMU achieves a higher generalization performance while greatly saving the communication overhead in most cases.

## 2. Related Works

**Compression-Efficient FL.** Extensive studies have been dedicated to reducing the amount of data required for gradient transmission and thus improving the communication efficiency. Using the unbiased compression method, QSGD (Alistarh et al., 2017), FedPAQ (Reisizadeh et al., 2020) and ECQ-SGD (Wu et al., 2018) compress the gradients uploaded to the server while keeping the original data integrity and expectation unchanged to save the communication cost. For biased compression, by leveraging the sign operation, signSGD (Bernstein et al., 2018a;b) can compress the gradients up to 1 bit. While the sparsification-based methods like TopK (Stich et al., 2018; Alistarh et al., 2018), which only keeps the largest $K$ gradients, is another communication-efficiently biased compression method. Other methods, like FedZip (Malekijoo et al., 2021) and Qsparse-local-SGD (Basu et al., 2019), incorporate both the quantization and sparsification. A direct application of biased compression, however, may lead to performance degradation and slower convergence rates due to the bias accumulation (Beznosikov et al., 2023). To address this, optimization techniques have been introduced to mitigate the negative effects of bias. For example, FedEF (Li & Li, 2023) and EF21 (Richtárik et al., 2021) employ error feedback, while MARINA (Gorbunov et al., 2021) and DIANA (Mishchenko et al., 2024) leverage the compression of gradient differences, both of which enhance the model performance and convergence speed. However, the performance of these algorithms is limited by the client participation rate. In this work, we adopt the sign operation to improve communication efficiency with partial participation of clients, which also helps enhance the generalization capability of the globally aggregated model as shown by Chen et al. (2024; 2021); Foret et al. (2020).

**Generalization-Enhanced FL.** In the advancement of FL algorithms, in parallel, various techniques have emerged to improve the generalization performance. By using momentum in FL, one can track the historical information of gradients, suppress the noise and reduce the instability of

*Table 1.* Summary of notations.

| | |
|---|---|
| $T, t$ | number, index of communication rounds |
| $K, k$ | number, index of local update step |
| $\eta, \gamma_1$ | local, global learning rate |
| $\beta_1, \beta_2$ | momentum coefficients |
| $\gamma_2$ | weight decay factor |
| $y_{t,k}^i$ | client $i$'s model at round $t$ and step $k$ |
| $x_t$ | aggregated server model after round $t$ |
| $\mathcal{M}, m$ | set of clients with cardinality $m$ |
| $\mathcal{N}_t, n$ | set of sampled active clients with cardinality $n$ |

model updates. Benefiting from this, methods such as MV-sto-signSGD-SIM (Sun et al., 2023) and FedAdam (Reddi et al., 2020) apply momentum instead of directly updating with gradients, while PR-SGD-Momentum (Yu et al., 2019) first updates the momentum and then combines the new gradient with a weight of the momentum. These methods enhance the model generalization and accelerate convergence in FL. In this work, we employ two sliding average functions to update momentum after calculating the new gradient, a technique introduced by Lion (Chen et al., 2024) to effectively store more historical gradient data. Also, since weight decay regularization has been shown to outperform $\ell_2$ regularization in preventing overfitting and enhancing generalization (Loshchilov, 2017), we leverage a weight decay strategy to mitigate the impact of data heterogeneity and further improve generalization performance. The work most closely related to ours is distributed Lion (Liu et al., 2024), which leverages the Lion optimizer to reduce communication overhead by extending it to the distributed setting with the full client participation and iid data. However, it lacks exploration of the partial participation and non-iid data scenarios, which are the major challenges brought by FL.

## 3. Proposed Method

### 3.1. Notations and Preliminaries

The general optimization problem of federated learning (FL) can be formulated as:

$$\min_{x \in \mathbb{R}^d} f(x) := \frac{1}{m} \sum_{i=1}^{m} F_i(x), \tag{1}$$

where $F_i(x) \triangleq \mathbb{E}_{\xi \sim D_i}[F_i(x, \xi)]$ represents the local loss function of the $i$-th client with the data sample $\xi$ drawn from distribution $D_i$. Under the FL settings, data is typically heterogeneous, implying that for two clients $i$ and $j$, the distributions $D_i$ and $D_j$ can be extremely different. Moreover, the FL systems often operate under a limited bandwidth, which renders the communication overhead associated with the exchange of model parameters a significant bottleneck.

Current approaches in FL often prioritize either mitigating data heterogeneity to enhance generalization or compress-

ing model updates to alleviate communication, rather than addressing both challenges concurrently. Specifically, most compression-based FL algorithms (Bernstein et al., 2018a;b; Li & Li, 2023; Wen et al., 2017) significantly reduce the communication cost, with a generalization performance typically comparable to or slightly lower than that of standard FedAvg (McMahan et al., 2017). On the other hand, most optimization-based FL strategies (Karimireddy et al., 2020), which involve exchange of full-precision model updates, and even additional control variables or informative representations, aim to mitigate the data heterogeneity issue, but at the cost of a huge communication overhead.

The recently proposed SCALLION algorithm (Huang et al., 2023) integrates the control variable-based SCAFFOLD framework with incremental variable compression methods, achieving a comparable performance with SCAFFOLD while substantially reducing the upload communication cost. Nonetheless, SCALLION additionally requires to double the download communication overhead for the transmission of control variables. CompressedScaffnew (Condat et al., 2022) and TAMUNA (Condat et al., 2023) also combine the control variates with the model compression. However, these methods rely on the permutation-based compression schemes, which are relatively complex and less flexible. LoCoDL (Condat et al., 2024) extends these two works by supporting a broader class of compressors and demonstrating a convergence acceleration in the convex problems, but it focuses exclusively on the convex setting. Additionally, FedComLoc (Yi et al., 2024) and Sparse-ProxSkip (Meinhardt et al., 2025) make attempts to explore the client drift under the non-convex objectives. However, FedComLoc's performance may degrade under the compressed communication due to its reliance on the communication variables, while Sparse-ProxSkip assumes the full client participation, which may not always be feasible in the real-world FL scenarios. These observation then impose a critical question for the field of compression-efficient FL: can we design an approach to effectively mitigate both the communication bottleneck and data heterogeneity simultaneously with partial participation of clients under the non-convex objectives?

### 3.2. Symbolizing Client Updates

Before answering this question, we revisit the standard FL framework, i.e., FedAvg (McMahan et al., 2017). With FedAvg, clients perform local training using their own datasets that are distributed over clients and non-iid in nature. The server then aggregates these locally trained models to update the global model, which subsequently serves as the initial model for the next round of training. However, due to the data heterogeneity, clients' model updates often differ in both the direction and magnitude. Consequently, when model updates from different clients with large deviations are averaged, some updates with relatively small magnitudes

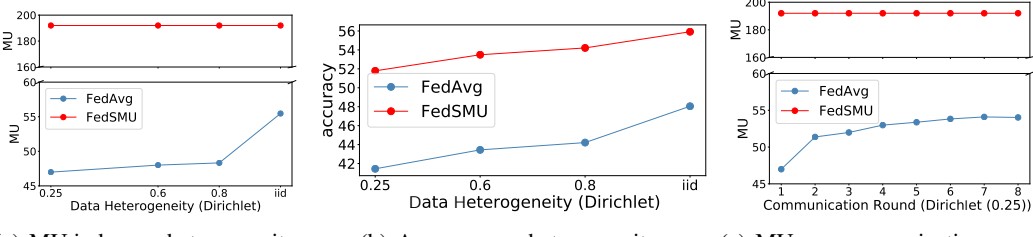

(a) MU index vs. heterogeneity     (b) Accuracy vs. heterogeneity     (c) MU vs. communication round

*Figure 1.* Magnitude uniformity (MU) index and top validation accuracy of FedAvg and FedSMU (ours) on CIFAR-100 with CNN model.

may be overlooked. For instance, we consider three clients, $i_1$, $i_2$ and $i_3$, whose model updates along one dimension are $+10$, $-1$, and $-1$, respectively. In this case, the updates have opposite directions, while the magnitude of client $i_1$'s update is much larger than those of clients $i_2$ and $i_3$. After averaging at the server (i.e., the global model's update becoming $+8/3$), the contribution of clients $i_2$ and $i_3$ to the global model's update will be ignored, since the update direction is now dominated by client $i_1$. Thus, a direct averaging may neglect contributions from the smaller updates and potentially compromise the fairness among clients.

To address this, and motivated by the Jain's fairness index (Jain et al., 1984), we propose a new metric called the Magnitude Uniformity (MU) index to reflect clients' contribution to the global model update. Through empirical analysis, we explore the relationship between this MU index and the local data heterogeneity, which in turn impacts the generalization performance of globally aggregated model.

**Definition 3.1.** (**Magnitude Uniformity**). We define the magnitude uniformity across $m$ clients at the $t$-th communication round as:

$$\Phi_t \triangleq \sum_{j=1}^{d} \frac{\left(\sum_{i \in \mathcal{M}} \hat{g}_t^{i,j}\right)^2}{\|\mathcal{M}\| \sum_{i \in \mathcal{M}} \left(\hat{g}_t^{i,j}\right)^2}, \; \hat{g}_t^{i,j} = \|y_{t,K}^{i,j} - y_{t,0}^{i,j}\|, \tag{2}$$

where $y_{t,K}^{i,j}$ denotes the $j$-th dimension ($d$ dimensions in total) of client $i$'s model at round $t$ and local step $K$, and $\hat{g}_t^{i,j}$ denotes the magnitude of client $i$'s model update in this dimension $j$ at round $t$. Similar to the Jain's fairness index, a higher value of the magnitude uniformity $\Phi_t$ indicates a more uniform contribution from the clients, thus suggesting a more balanced representation of the clients' data in the global model. Theoretically, such a uniformity may lead to a global model better capturing the information from all the local clients. Consequently, one may raise a question: is this magnitude uniformity index affected by the data heterogeneity across locally distributed clients, and does it further influence the global model's generalization performance?

Seeking for an answer to this question, we empirically examine the correlation between this Magnitude Uniformity index and the global model's generalization performance

under varying data heterogeneity on the CIFAR-100 dataset. The experiment involves 100 clients with a partial participation rate of 10%. As observed from Figures 1(a) and 1(b), for FedAvg, an increase in the data heterogeneity leads to a decrease in the Magnitude Uniformity index, accompanied by a deterioration in the generalization performance. This suggests that with FedAvg, data heterogeneity leads to a significant difference in the magnitude of model updates across clients, resulting in an unstable global aggregation and poorer generalization performance. Additionally, as shown in Figure 1(c), the Magnitude Uniformity index tends to rise during the FedAvg training, suggesting that the early stage of an FL system forces a gradual narrowing on the magnitude difference of model updates across clients.

A straightforward approach to enhance the Magnitude Uniformity index for FL is to apply a sign operation to the local clients' updates, ensuring that model updates have the uniform magnitude from all the clients. Specifically, after this sign operation, the local model updates for the three clients $i_1$, $i_2$ and $i_3$ in the previous example become $+1$, $-1$, and $-1$, respectively. This process guarantees that each client's model update contributes equally to the globally aggregated model, thereby reducing the impact of model heterogeneity and promoting fairness. By converting the magnitudes of model updates into their respective signs, we actually emphasize the directions of their updates rather than the magnitudes, which could help balance the contributions from different clients' model updates and lead to a more representative and informative global model. Moreover, by symbolizing the updates we can reduce the communication cost to 1 bit per dimension, offering a potential solution to enhancing generalization while saving the communication.

In fact, numerous sign-based compression methods (Bernstein et al., 2018a;b; Wen et al., 2017) have been applied in FL. While theoretically performant, their empirical results often show only marginal improvements or comparable performance to FedAvg. Thus, effectively leveraging the sign operation to simultaneously mitigate the communication overhead and enhance generalization in federated learning remains a challenging and unresolved issue. On the other hand, many optimization techniques have been proposed to improve generalization for the centralized learning, such as momentum, Adam, and weight decay. A brute force ap-

proach could be directly incorporating the sign operations with these optimization strategies in FL, formulating the algorithm design as a program search to identify federated optimization algorithms that can incorporate sign compression. However, this approach is computationally expensive.

Fortunately, in the context of centralized learning, Lion (EvoLved Sign Momentum) optimizer (Chen et al., 2024) employs the sign operation to compute the updates while tracking momentum, which has demonstrated an overall outstanding performance across various models and tasks. Compared to the simple signSGD (Bernstein et al., 2018a;b), Lion leverages the dual momentum tracking and weight decay, significantly improving generalization ability of the trained models. Inspired with our observation on the impact of Magnitude Uniformity index on FL's generalization performance, and further motivated by the success of Lion in centralized learning, we thus propose a new federated optimization algorithm aiming at both reducing the communication overhead and enhancing the generalization performance, through symbolizing the client model updates.

### 3.3. Proposed FedSMU

To leverage the structured design of Lion optimizer and minimize the communication overhead, we propose our FedSMU algorithm for federated learning, which splits the Lion optimizer's framework of momentum tracking and weight decay to be carried out independently at the server and each client, respectively, as summarized in Algorithm 1.

Specifically, at each communication round $t$, our proposed FedSMU implements the following steps:

1. Participating clients initialize their local models, denoted as $y_{t,0}^i$, based on the current global model $x_t$.

2. Each client conducts $K$ steps of local stochastic gradient descent (SGD) to compute the model update $g_t^i$.

3. Each client symbolically represents its model updates by using the momentum and the sign operations.

4. The server receives and aggregates these symbolic updates, denoted as $u_t^i$, to update the global model $x_{t+1}$ by incorporating the weight decay.

Such a design offers two significant advantages to our FedSMU algorithm. First, it fully leverages the structure of the Lion optimizer, thereby enhancing the generalization performance of the global model. It is also worth noting that in the special case where the number of local update steps and clients are set to 1, our optimizer essentially reverts to the standard Lion. Second, by transmitting only 1-bit update for each dimension of the model parameters between the clients and server, we substantially reduce the communication overhead in the FL systems.

We also notice that inspired by advantages of the Lion optimizer in the centralized learning, there has been other works, e.g., FedLion proposed by Tang & Chang (2024), incorporating Lion into the local updates of federated learning. However, FedLion simply uses the vanilla Lion algorithm for the local updates to replace SGD, resulting in a communication cost that is even significantly higher than that of FedAvg, as the extra momentum terms need to be transmitted. Compared to FedLion, our FedSMU out-stands as follows. **1) Effective utilization of Lion framework.** FedSMU divides the execution of Lion optimizer across the clients and server. In contrast, FedLion merely executes the Lion algorithm locally in parallel as a local optimization strategy, failing to exploit the complete structure of Lion. **2) Communication cost saving.** In addition to the model updates, FedLion requires an additional transmission of the full-precision momentum terms, resulting in a significantly higher communication cost compared to FedSMU, which only necessitates a 1-bit communication for each dimension of model updates. This substantial reduction in communication overhead is another key advantage of our FedSMU.

## 4. Theoretical Results on Convergence

We now present the convergence analysis of our proposed FedSMU for the general non-convex functions. In general, our analysis is based on the following three standard assumptions, which are commonly satisfied by a range of non-convex objective functions.

**Assumption 4.1.** (Lipschitz Gradient). *For all $i \in \mathcal{M}$, the function $F_i(x)$ is L-smooth: $||\nabla F_i(x) - \nabla F_i(y)|| \leq L||x - y||$ for all $x, y \in \mathbb{R}^d$.*

**Assumption 4.2.** (Bounded Variance). *For all $i \in \mathcal{M}$, the function $F_i(x, \xi)$ has a locally-bounded variance $\sigma_l^2$: $\mathbb{E}[||\nabla F_i(x, \xi) - \nabla F_i(x)||]^2 \leq \sigma_l^2$ for all $x \in \mathbb{R}^d$.*

**Assumption 4.3.** (Bounded Gradients). *For all $i \in \mathcal{M}$, the function $F_i(x, \xi)$ has a bounded gradient: $||\nabla F_i(x, \xi)|| \leq G$ for all $x \in \mathbb{R}^d$.*

For the non-convex optimization problem, Assumptions 4.1 and 4.2 are standard and widely adopted in various literature of FL (Reddi et al., 2020; Bottou et al., 2018; Reddi et al., 2016; Ghadimi & Lan, 2013; Li & Orabona, 2019). Assumption 4.3 is commonly used in the convergence analysis of sign-based methods, such as the distributed signSGD (Sun et al., 2023; Jin et al., 2020).

**Theorem 4.4.** *Under Assumptions 4.1, 4.2, and 4.3, when $0 < \eta \leq \frac{1}{4LK}$, $\gamma_1 = \mathcal{O}(\frac{1}{L\sqrt{T}})$ and $1 - \beta_1 = \mathcal{O}(\frac{1}{\sqrt{T}})$, we*

**Algorithm 1** Federated learning through Symbolic Model Updates (FedSMU) algorithm.

---

**Server Initialization**: $x_1$;
**Client Initialization**: $m_0^i = 0$;
**for** *each round $t = 1, 2, ...T$* **do**
    sample clients $\mathcal{N}_t \subseteq \mathcal{M}$
    **for** *each client $i \in \mathcal{N}_t$ in parallel* **do**
        receive and initialize local model $y_{t,0}^i = x_t$
        **for** *each local step $k = 1, 2, \ldots, K$* **do**
            $y_{t,k}^i = y_{t,k-1}^i - \eta \nabla F_i(y_{t,k-1}^i, \xi_{t,k-1}^i)$
        **end**
        $g_t^i = y_{t,K}^i - y_{t,0}^i$
        $u_t^i = \text{Sign}(\beta_1 m_{t-1}^i + (1 - \beta_1) g_t^i)$
        $m_t^i = \beta_2 m_{t-1}^i + (1 - \beta_2) g_t^i$ (for $i \notin \mathcal{N}_t$, $m_t^i = m_{t-1}^i$)
        send $u_t^i$ to server
    **end**
    // at server:
    $x_{t+1} = x_t + \gamma_1 (\frac{1}{n} \sum_{i=1}^n u_t^i - \gamma_2 x_t)$
    broadcast $x_{t+1}$
**end**

---

*have:*

$$\Psi \leq \frac{L(f(x_0) - \min f)}{\sqrt{T}} + \frac{3G\sqrt{d}\phi}{nT(1 - \beta_2)} + \frac{6\eta d\tau_{max}}{LT(1 - \beta_2)}$$
$$+ \frac{12\eta}{T}\sqrt{\frac{d(2K\sigma_l^2 + 4K^2\sigma_l^2 + 4K^2G^2)}{1 - \beta_2}}$$
$$+ \frac{6Gd}{\sqrt{n}} + \frac{2d}{\sqrt{T}},$$

$$\tag{3}$$

*where $\Psi = \frac{1}{T}\sum_{t=1}^T \mathbb{E}[||\nabla f(x_t)||_1]$, $\phi = \sum_{i=1}^m \|\frac{1}{G}\nabla F_i(x_0)\|$, $d$ denotes the dimensions of parameters, $\tau_{max} = \max\{\tau^i\}_{1 \leq i \leq m, 1 \leq t \leq T}$ and $\tau^i$ denotes client $i$'s participation interval.*

*Proof.* See Appendix B for the detailed proof. $\square$

*Remark* 4.5. The convergence rate of our FedSMU is $\mathcal{O}(\frac{1}{\sqrt{T}})$ when $T$ is sufficiently large, matching with the convergence rates of existing FL algorithms, such as Fe-dAvg and FedPAQ (Reisizadeh et al., 2020). Note that $\tau_{max}$ represents the maximum participation interval among all the clients, indicating that larger participation intervals result in a slower convergence. Note that $d$ represents the model dimension and directly influences the rate of convergence, i.e., a larger model dimension results in a slower convergence. Increasing the number of workers $n$ leads to a tighter error bound. Further in Appendix C.5, we show that though a higher precision quantization can reduce the quantization error, it may slow down the overall convergence rate in some cases. We also verify the relationship between the

convergence speed and $\tau_{max}$ and $d$ through experiments in Appendix C.6.

*Remark* 4.6. The original work of Lion (Chen et al., 2024) does not include a convergence analysis. Our theoretical analysis provides the relevant convergence rate for the Lion optimizer. Specifically, by setting $n = 1, \tau_{max} = 1$ and $K = 1$, the convergence rate of our FedSMU reduces to that of the Lion optimizer. However, FedLion (Tang & Chang, 2024) cannot be reduced to a standard Lion optimizer, because it merely parallelizes the execution of Lion optimizer at the client side and incorporates multi-precision quantization for communication compression.

## 5. Experiments

We conduct comprehensive comparative experiments to validate the superior performance of FedSMU in scenarios involving different partial participation rates and data heterogeneity degrees. The additional experiments across more scenarios and ablation studies are provided in Appendix C, and the implementable code of our proposed FedSMU algorithm is available at https://github.com/lxy66888/fedsmu.git.

### 5.1. Experimental Setup

**Models and Dataset.** We evaluate our FedSMU and the other baseline algorithms on three real-world visual and language datasets: CIFAR-10, CIFAR-100 (Krizhevsky et al., 2009) and neural machine translation on Shakespeare, with the same train/test splits as in (Acar et al., 2021). Each client is assigned an uncertain number of classes, and the data within each class varies widely, with the labels of client samples generated according to a Dirichlet distribution. For instance, using Dirichlet-0.25 on CIFAR-10, there are approximately 80% of each client's samples belonging to around three or four different classes. We employ the CNN model with LeNet architecture and RNN model both similar to previous studies (McMahan et al., 2017). Furthermore, to demonstrate the applicability of our method to more complex models and datasets, we evaluate the performance of different FL algorithms using a larger model, ResNet18 (He et al., 2016) and ViT-S (Dosovitskiy et al., 2021), and a more challenging dataset, Tiny-ImageNet, a reduced version of the ILSVRC (ImageNet Large Scale Visual Recognition Challenge) (Russakovsky et al., 2015) classification dataset. For additional details on the experimental setup, please refer to Appendix A.

**Comparison Algorithms.** We compare the validation (test) performance of our FedSMU with several other baselines, including the optimization-based FL algorithms such as Fe-dAvg (McMahan et al., 2017), FedLion (Tang & Chang, 2024), and SCAFFOLD (Karimireddy et al., 2020), as well as the compression-based FL algorithms such as FedEF-HS

*Table 2.* Performance comparison under various settings, where a smaller Dirichlet parameter indicates a higher data heterogeneity, and L and H indicate low and high participation rates, respectively. For CIFAR-10 and CIFAR-100, a LeNet model is used, and for Shakespeare, an RNN model is employed. Bold numbers indicate the best performance.

| Dataset | Setting | FedAvg | SCAFFOLD | SCALLION | FedEF-HS | FedEF-TopK | FedEF-Sign | FedLion | FedSMU |
|---|---|---|---|---|---|---|---|---|---|
| | | | | Top-1 Test Accuracy (%). | | | | | |
| CIFAR-100 | Dir (0.25)-L | 41.44 | 41.28 | 42.68 | 38.31 | 44.29 | 37.79 | 45.09 | **51.87** |
| | Dir (0.6)-L | 41.36 | 45.04 | 43.28 | 38.63 | 44.41 | 40.34 | 47.19 | **53.79** |
| | Dir (0.25)-H | 42.29 | 50.49 | 45.24 | 37.03 | 42.69 | 36.12 | 48.33 | **52.35** |
| | Dir (0.6)-H | 43.44 | 50.02 | 45.84 | 36.09 | 42.72 | 38.99 | 48.85 | **54.2** |
| CIFAR-10 | Dir (0.25)-L | 80.95 | **81.6** | 80.91 | 78.35 | 80.11 | 77.87 | 79.04 | 80.12 |
| | Dir (0.6)-L | 82.42 | 82.36 | 81.18 | 79.29 | 81.73 | 79.68 | 80.94 | **82.48** |
| | Dir (0.25)-H | 80.6 | **83.31** | 81.42 | 78.17 | 79.92 | 78.34 | 81.61 | 80.74 |
| | Dir (0.6)-H | 81.43 | **84.12** | 81.75 | 78.75 | 81.42 | 79.38 | 83.15 | 82.66 |
| Shakespeare | noniid-H | 47.58 | **51.28** | 47.86 | 45.79 | 46.21 | 45.00 | 47.11 | 47.81 |

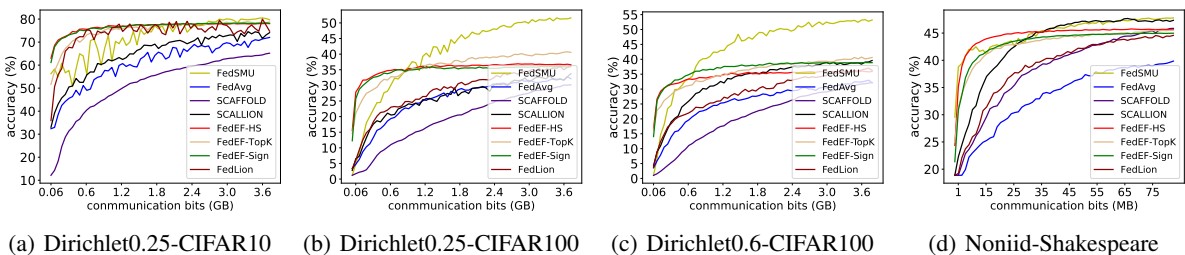

| (a) Dirichlet0.25-CIFAR10 | (b) Dirichlet0.25-CIFAR100 | (c) Dirichlet0.6-CIFAR100 | (d) Noniid-Shakespeare |
|---|---|---|---|

*Figure 2.* Convergence performance vs. number of uplink communication bits on CIFAR-10, CIFAR-100 and Shakespeare, with 100 clients and 10% participation. For CIFAR-10 and CIFAR-100, a LeNet model is used, and for Shakespeare, an RNN model is employed.

(Li & Li, 2023), FedEF-TopK (Li & Li, 2023), FedEF-Sign (Li & Li, 2023), and SCALLION (Huang et al., 2023). It is worth noting that FedLion (Tang & Chang, 2024) involves a parallel execution of the Lion optimizer on the local clients, requiring the upload of full-precision momentum updates in addition to the compressed model updates. Consequently, the communication overhead of FedLion is higher than Fe-dAvg, even when the model updates are compressed. Additionally, SCAFFOLD needs twice of the communication cost compared to FedAvg. Though SCALLION uploads the compressed incremental updates, it still results in doubling the communication overhead during the download phase. Since the distributed Lion (Liu et al., 2024) can only be applied under the full client participation, we include a comparison with it in Appendix C.9, which also demonstrates the advantages of our FedSMU.

**Implementation.** We evaluate the performance of the global model on the CIFAR-10, CIFAR-100 and Shakespeare datasets, by utilizing 100 clients with high (H) and low (L) client participation rates of 10% and 3%, respectively. For the ResNet18 and ViT-S model, we adopt the client number of 10 and the participation rate of 30%. Clients are uniformly sampled at random without replacement at each round. The learning rates and hyperparameters for all approaches are individually tuned via a grid search. For additional details on hyperparameter settings, please refer

to Appendix A.

## 5.2. Experimental Results

### 5.2.1. PERFORMANCE EVALUATION

Experimental results for all the comparison methods under three datasets are shown in Table 2 and Figure 2. In most cases, our FedSMU demonstrates a superior performance compared to the other baselines (especially compression-based methods) with varying data distributions and client participation rates. The results effectively demonstrate that our FedSMU performs well on both image classification and text prediction tasks. We attribute this improvement to our design, which mimics the Lion optimizer and incorporates symbolic updates, momentum tracking, and weight decay. In contrast, other compression-based methods, such as TopK and group sign employed by FedEF-TopK and FedEF-Sign, compress the communication traffic but consistently exhibit a poorer generalization performance.

Note that our FedSMU generally presents a more significant performance gain on CIFAR-100 for image classification. For CIFAR-10, though our FedSMU outperforms the compression-based FL algorithms, it is still less effective than the optimization-based algorithms, such as FedAvg and SCAFFOLD. Here, we discuss about the possible reason for this slight degradation on CIFAR-10. In a federated hetero-

*Table 3.* Performance comparison under various settings with the ResNet18 network model. Bold numbers indicate the best performance.

| | | Top-1 Test Accuracy (%). | | | | | | | |
|---|---|---|---|---|---|---|---|---|---|
| Dataset | Setting | FedAvg | SCAFFOLD | SCALLION | FedEF-HS | FedEF-TopK | FedEF-Sign | FedLion | FedSMU |
| CIFAR-10 | Dir (0.25) | 81.74 | **85.62** | 83.75 | 80.43 | 82.54 | 83.93 | 82.44 | 83.54 |
| CIFAR-100 | Dir (0.25) | 47.41 | 48.76 | 48.15 | 47.24 | 48.07 | 49.03 | 43.75 | **49.76** |
| Tiny-ImageNet | Dir (0.25) | 29.79 | 33.35 | 31.91 | 31.68 | 31.05 | 31.17 | 28.36 | **33.53** |

*Table 4.* Performance comparison under various settings with the ViT-S network model. Bold numbers indicate the best performance.

| | | Top-1 Test Accuracy (%). | | | | | | | |
|---|---|---|---|---|---|---|---|---|---|
| Dataset | Setting | FedAvg | SCAFFOLD | SCALLION | FedEF-HS | FedEF-TopK | FedEF-Sign | FedLion | FedSMU |
| CIFAR-10 | Dir (0.25) | 74.24 | 76.26 | 75.12 | 74.11 | 74.15 | 74.08 | 71.75 | **76.71** |
| CIFAR-100 | Dir (0.25) | 41.72 | 48.27 | 46.37 | 51.79 | 45.28 | 44.02 | 43.66 | **53.24** |

geneous scenario involving CIFAR-100, which comprises 100 categories as compared to 10 categories for CIFAR-10, each client typically handles a subset of 13-16 (or 20-25) categories when setting Dirichlet-0.25 (or Dirichlet-0.6). Thus, with such a high degree of heterogeneity incurred on CIFAR-100, the model updates from clients are more deviated, allowing our FedSMU to be more effective and demonstrate a more significant improvement than on CIFAR-10.

To confirm that our algorithm can maintain a good performance in larger network models, we also conduct evaluations on the ResNet18 and ViT-S models. Due to limited computing resources, we set 10 clients in total with a participation rate of 30%, and show results in Table 3 and Table 4. It can be observed that FedSMU outperforms most baseline methods with ResNet18 and ViT-S models. Though it remains slightly inferior to SCAFFOLD on the CIFAR-10 dataset with ResNet18 model, it achieves a superior performance with ViT-S model, further demonstrating its greater advantages in complex network architectures.

To further evaluate the generalization performance, we define generalization as the test accuracy that an algorithm can achieve at the same level of training accuracy. We show that FedSMU continues to demonstrate the best generalization performance, with detailed analysis given in Appendix C.1.

### 5.2.2. GENERALIZATION VS. PARTICIPATION RATE

We then evaluate the effect of different participation rates on all the algorithms, while keeping the number of participating clients consistent at each communication round. Results in Table 5 indicate that FedSMU achieves the highest accuracy in most cases. Specifically, when the number of participating clients is maintained at 10, and when the participation rate decreases from 0.2 to 0.05, FedAvg (McMahan et al., 2017), SCAFFOLD (Karimireddy et al., 2020), SCALLION (Huang et al., 2023), FedEF-HS, FedEF-TopK and FedEF-Sign (Li & Li, 2023)would experience a severe performance

*Table 5.* Top validation accuracy (%) under different participation rate with Dirichlet-0.25 on CIFAR-100 dataset and LeNet model, where NTC indicates the number of total clients, and PR indicates the participation rate. Bold numbers indicate the best performance.

| NTC / PR | 50 / 0.2 | 100 / 0.1 | 150 / 0.066 | 200 / 0.05 |
|---|---|---|---|---|
| FedSMU | **52.39** | **52.35** | **51.75** | **50.22** |
| FedAvg | 46.62 | 42.29 | 40.93 | 39.44 |
| SCAFFOLD | 52.52 | 50.49 | 39.39 | 37.31 |
| SCALLION | 48.07 | 45.24 | 36.54 | 35.49 |
| FedEF-HS | 42.68 | 37.03 | 34.37 | 31.71 |
| FedEF-TopK | 47.25 | 42.69 | 40.23 | 37.41 |
| FedEF-Sign | 42.68 | 36.12 | 33.94 | 31.04 |
| FedLion | 48.41 | 48.33 | 47.81 | 48.74 |

deterioration of 7.18%, 15.21%, 12.58%, 10.97%, 9.84% and 11.64%, respectively. In contrast, FedSMU maintains a more stable and superior performance, with only a 2.17% deterioration. These results indicate that our algorithm is minimally impacted by client participation rates and demonstrates greater stability under partial client participation. We attribute this to the use of symbolic operations for the client updates, which effectively leverages each client's update even at very low participation rates. Specifically, when the client participation rate is low, data heterogeneity may cause the update of certain clients to dominate due to larger magnitudes. Symbolic operations can mitigate this by normalizing the update amplitudes, ensuring that the contributions of all clients are fully considered.

### 5.2.3. GENERALIZATION VS. DATA HETEROGENEITY

We further study the influence of data heterogeneity on the generalization performance of our FedSMU vs. FedAvg and SCAFFOLD. From Table 6, it is evident that FedSMU outperforms the other two algorithms. By computing the top accuracy difference between the iid and Dirichlet-0.25 settings in Table 6, we observe a degradation of 3.57%,

*Table 6.* Top validation accuracy (%) under different data heterogeneity with 100 clients and 10% participation rate on CIFAR-100 dataset and LeNet model, where Dirichlet-0.25 indicates the highest heterogeneity and iid indicates the lowest heterogeneity.

| Algorithm | Dirichlet-0.25 | Dirichlet-0.6 | Dirichlet-0.8 | iid |
|---|---|---|---|---|
| FedSMU | **52.35** | **54.2** | **54.92** | **55.92** |
| FedAvg | 41.44 | 43.44 | 44.21 | 48.05 |
| SCAFFOLD | 50.49 | 50.02 | 53.89 | 54.78 |

6.61% and 4.29% in the top accuracy for FedSMU, FedAvg and SCAFFOLD, respectively. Thus, FedSMU is affected less significantly by the degree of data heterogeneity.

Besides, with a horizontal comparison, the improvement of FedSMU over FedAvg is 10.91%, 10.76%, 10.71% and 7.87% for Dirichlet-0.25, Dirichlet-0.6, Dirichlet-0.8, and iid distributions, respectively. This indicates that FedSMU achieves a higher performance gain with the increasing degree of data heterogeneity. These results also validate that in highly heterogeneous data scenarios, where the difference between clients' model updates becomes greater, FedSMU can alleviate the local model heterogeneity through symbolic updates. This promotes the aggregation stability and improves generalization performance of the global model.

### 5.2.4. LIMITATIONS

Though our FedSMU effectively enhances the generalization performance while reducing the communication overhead, it may still have some limitations. First, our compression relies on the fixed-precision symbol quantization, which might not be optimal for the adaptive scenarios. Exploring adaptive bit quantization further in our future inresearch is promising to address this limitation. Second, due to the partial participation inherent in federated learning, the server must broadcast the new global model to initialize the newly participating clients at each communication round. This constraint prevents the direct application of our compression techniques to the downloaded global model in FedSMU. We will consider some model lightweight techniques, such as mixed-precision model compression, as a promising future research to compress the server-to-client communication in our FedSMU algorithm. Third, though our FedSMU reduces communication costs, it does not necessarily offer an advantage in reducing the communication rounds. This may be due to the noise introduced by the sign operation, which enhances the model's generalization but meanwhile slows down its convergence. From a theoretical perspective, though the generalization properties (Venkateswaran et al., 2023) of FedAvg under various assumptions have been extensively examined, such guarantees for the compression-based FL approaches remain an open problem. Last, our current focus is only on symbolized local updates with SGD optimizer. However, integrating adaptive optimizers like AdamW to replace SGD (Douillard

et al., 2024) may further enhance the performance on the modern large language models.

## 6. Conclusion

In this paper, we have proposed the FedSMU algorithm that could effectively alleviate both the communication cost and data heterogeneity issues of federated learning. The key design was the symbolization of local client updates which were introduced to balance the contribution of each client and avoid the dominance by some relatively large update values. We carried out theoretical convergence analysis, and empirically showed that FedSMU converged faster to a higher top accuracy under the same communication cost. Under the condition of a very small partial client participation rate and relatively high data heterogeneity, FedSMU still demonstrated a better performance.

## Acknowledgements

This work was supported in part by the National Natural Science Foundation of China under Grants 62320106003, U24A20251, 62401357, 62401366, 62431017, 62125109, 62371288, 62301299, 62120106007, in part by the Program of Shanghai Science and Technology Innovation Project under Grant 24BC3200800, and in part by the AI Laboratory of United Automotive Electronic Systems (UAES) Co. under Grant 2025-3270.

## Impact Statement

Federated learning is a crucial machine learning framework that prioritizes the privacy protection. Our work proposes a novel federated compression algorithm aiming at simultaneously improving the communication efficiency and generalization performance. This approach can be beneficial for the privacy-sensitive domains, such as healthcare and finance, since transmitting only the sign information makes it more difficult for the attackers to reconstruct the original data, thereby enhancing the privacy protection level. In addition, by effectively reducing the communication overhead, this work may also enable the practical deployment of federated learning in some extreme scenarios, such as the low-bandwidth networks.

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

# Appendix

## A. Detailed Experimental Setup

We utilize the visual datasets including CIFAR-10, CIFAR-100 and Tiny-ImageNet. CIFAR-10 and CIFAR-100 are two classic image classification datasets created by the Canadian Institute for Advanced Research (CIFAR). The CIFAR-10 dataset consists of 10 classes of images, with each class containing 6000 32x32-pixel color images. The CIFAR-100 dataset is an extension of CIFAR-10, containing 100 classes of images. These 100 classes are divided into 20 superclasses, each containing 5 subclasses. Each subclass contains $600$ $32 \times 32$-pixel color images. Both of them comprise 50,000 images for training and 10,000 images for testing. The Tiny-ImageNet dataset is a reduced version of the ILSVRC classification dataset. It consists of 200 distinct object categories with $64 \times 64$-pixel color images of 3 channels. Each category includes 500 images for training and 50 for testing.

For CIFAR-10 and CIFAR-100, we employ a LeNet model comprising two convolutional layers with sixty four $5 \times 5$ filters, two $2 \times 2$ max pooling layers, two fully connected layers with 384 and 192 neurons, and a softmax layer. We also use a larger network, ResNet18, to confirm that our algorithm still performs well in a larger network. ResNet18 contains 16 convolutional layers. These convolutional layers are distributed across several residual blocks, each containing two $3 \times 3$ convolutional layers. Additionally, there is a $7 \times 7$ convolutional layer at the beginning of the network. At the end of the network, there is a fully connected layer for output. Vision Transformer (ViT) adapts the Transformer architecture from natural language processing to image classification tasks. In this work, we employ the ViT-Small (ViT-S) variant, which incorporates image patching, positional encoding, and a 12-layer Transformer structure.

For Shakespeare dataset, we employ an RNN model. It consists of the input layer (receiving the input at the current time step), hidden layer (receiving the hidden state from the previous time step along with the input at the current time step to compute the new hidden state) and output layer (outputting a result based on the current state of the hidden layer).

All approaches are implemented in PyTorch 1.4.0 and CUDA 9.2, with GEFORCE GTX 1080 Ti throughout our experiments.

In most federated learning scenarios, the total number of clients is set to 100 with the participation rate of 0.1, which is a classical experimental setting, like what FedLion (Tang & Chang, 2024) and FedAvg(McMahan et al., 2017) do. Therefore, with the LeNet model, we set 100 clients with participation rates of 0.1 and 0.03 to verify the performance of our algorithm.

However, our computing resources (i.e., GEFORCE GTX 1080 Ti) are insufficient to support an experimental setting of 100 clients on the larger network model, we thus can reduce the number to 10 clients and set the participation rate to 0.3.

We tune the hyper-parameter over a grid to compare the performance of different methods. For local update in all methods, we tune the local learning rate over $\{1, 0.1, 0.01, 0.001\}$ and set up 5 epochs of local updates with the minibatch $B = 50$.

For our proposed method FedSMU, we tune the parameter $\beta_1$ and $\beta_2$ over $\{0.9, 0.99, 0.999\}$, respectively, and set them both to 0.9 for CIFAR-10, CIFAR-100 and Tiny-ImageNet, and 0.95 for Shakespeare. We tune the parameter $\gamma_1$ and $\gamma_2$ over $\{1, 0.1, 0.02, 0.018, 0.015, 0.013, 0.01, 0.005, 0.001\}$, respectively, since they are so sensitive, and set them to 0.015, 0.01 for CIFAR-10, 0.018, 0.01 for CIFAR-100, 0.01, 0.01 for Tiny-ImageNet, and 0.03, 0.01 for Shakespeare.

For FedSMUMC, we tune the parameter $\beta_1$ and $\beta_2$ over $\{0.9, 0.99, 0.999\}$, respectively, and set them both to 0.9 for CIFAR-100. We tune the parameter $\gamma_1$ and $\gamma_2$ over $\{1, 0.1, 0.01, 0.001\}$, respectively, and set them both 0.01 for CIFAR-100.

For Fed-LocalLion and Fed-GlobalLion, we tune the parameter $\beta_1$ and $\beta_2$ over $\{0.9, 0.99, 0.999\}$, respectively, and set them to 0.9 and 0.99 for CIFAR-100. We tune the parameter $\gamma_1$ and $\gamma_2$ over $\{1, 0.1, 0.01, 0.001\}$, respectively, and set them to 0.001, 0.01 for CIFAR-100. We tune the parameter $\eta_g$ in Fed-LocalLion over $\{1, 0.1, 0.01, 0.001\}$ and set them to 1 for CIFAR-100.

## B. A Proof of Theorem 4.4

*Proof.* We set $\Delta_t = \frac{1}{n}\sum_{i=1}^{n} u_t^i = \frac{1}{n}\sum_{i=1}^{n} \mathrm{Sign}[\beta_1 m_{t-1}^i + (1 - \beta_1)g_t^i]$ and $||\Delta_t||^2 = \sum_{j=1}^{d} |\Delta_t^j|^2$, where $d$ is the dimensions of parameters.

Since $\gamma_2$ is adjustable, so for each coordinate $j$, we can assume $|\gamma_2 x_t^j| \leq 1$ ($||\gamma_2 x||_\infty \leq 1$). It has been clarified by Chen et al. (2023) in the Abstract that "Lion is a theoretically novel and principled approach for minimizing a general loss function f(x) while enforcing a bound constraint $||x||_\infty \leq \frac{1}{\gamma_2}$." Here $\gamma_2$ is the weight decay coefficient and $x_t$ is the model.

Such an assumption has also been used in another algorithm (Liu et al., 2024) based on Lion optimizer. Thus we have $|\Delta_t^j - \gamma_2 x_t^j| \leq |\Delta_t^j| + |\gamma_2 x_t^j| \leq 2$, and then $||\Delta_t||^2 \leq d$ and $||\Delta_t - \gamma_2 x_t||^2 \leq 4d$.

With Assumption 4.1, we have

$$
\begin{aligned}
f(x_{t+1}) &\leq f(x_t) + \langle \nabla f(x_t), x_{t+1} - x_t \rangle + \frac{L}{2}||x_{t+1} - x_t||^2 \\
&= f(x_t) + \langle \nabla f(x_t), \gamma_1 \Delta_t - \gamma_1 \gamma_2 x_t \rangle + \frac{L}{2}||\gamma_1 \Delta_t - \gamma_1 \gamma_2 x_t||^2 \\
&= f(x_t) - \langle \nabla f(x_t), \gamma_1 \operatorname{Sign}(\nabla f(x_t)) \rangle + \langle \nabla f(x_t), \gamma_1 \Delta_t - \gamma_1 \gamma_2 x_t + \gamma_1 \operatorname{Sign}(\nabla f(x_t)) \rangle \\
&\quad + \frac{L}{2}||\gamma_1 \Delta_t - \gamma_1 \gamma_2 x_t||^2 \\
&= f(x_t) - \gamma_1 ||\nabla f(x_t)||_1 + \gamma_1 \underbrace{\langle \nabla f(x_t), \Delta^t - \gamma_2 x_t + \operatorname{Sign}(\nabla f(x_t)) \rangle}_{A} + 2L\gamma_1^2 d.
\end{aligned}
\tag{4}
$$

Considering the calculation of $A$:

$$
\begin{aligned}
A &= \langle \nabla f(x_t), \Delta_t - \gamma_2 x_t + \operatorname{Sign}(\nabla f(x_t)) \rangle \\
&= \langle \nabla f(x_t), \frac{1}{n}\sum_{i=1}^n u_t^i - \gamma_2 x_t + \operatorname{Sign}(\nabla f(x_t)) \rangle.
\end{aligned}
\tag{5}
$$

For any dimension $j$, assume $|\gamma_2 x_t^j| < 1$, and with Assumption 4.3, then we have $\nabla f(x_t^j)(\frac{1}{n}\sum_{i=1}^n u_t^{i,j} - \gamma_2 x_t^j + \operatorname{Sign}(\nabla f(x_t^j))) \leq 3|\nabla f(x_t^j)| = 3G|\frac{1}{G}\nabla f(x_t^j)| < 3G|\operatorname{Sign}(\nabla f(x_t^j))| < 3G|\frac{\gamma_1 \eta}{G}\nabla f(x_t^j) + \operatorname{Sign}(\nabla f(x_t^j))|$.

So,

$$
\begin{aligned}
A &< 3G||\frac{\gamma_1 \eta}{G}\nabla f(x_t) + \operatorname{Sign}(\nabla f(x_t))||_1 \\
&\leq 3\sqrt{d}G||\frac{\gamma_1 \eta}{G}\nabla f(x_t) + \operatorname{Sign}(\nabla f(x_t))||.
\end{aligned}
\tag{6}
$$

Substitute Eq. (6) into Eq. (4), we further have

$$
\begin{aligned}
f(x_{t+1}) - f(x_t) &\leq -\gamma_1||\nabla f(x_t)||_1 + \gamma_1 \underbrace{\langle \nabla f(x_t), \Delta_t - \gamma_2 x_t + \operatorname{Sign}(\nabla f(x_t)) \rangle}_{A} + 2L\gamma_1^2 d \\
&\leq -\gamma_1||\nabla f(x_t)||_1 + 3G\gamma_1\sqrt{d}\underbrace{||\frac{\gamma_1 \eta}{G}\nabla f(x_t) + \operatorname{Sign}(\nabla f(x_t))||}_{B} + 2L\gamma_1^2 d.
\end{aligned}
\tag{7}
$$

Taking the expectation of $B$, with Assumption 4.3 we have

$$
\begin{aligned}
\mathbb{E}(B) &\leq \mathbb{E}(\underbrace{||\frac{\gamma_1 \eta}{G}\nabla f(x_t) + \frac{\gamma_1}{nKG}\sum_{i=1}^n v_t^i||}_{\epsilon_t}) + \mathbb{E}(||\operatorname{Sign}(\nabla f(x_t)) - \frac{\gamma_1}{nKG}\sum_{i=1}^n v_t^i||) \\
&\leq \mathbb{E}(||\epsilon_t||) + \sqrt{\mathbb{E}(\frac{1}{n^2}\sum_{i=1}^n||\operatorname{Sign}(\nabla f(x_t)) - \frac{\gamma_1}{KG}v_t^i||^2)} \\
&= \mathbb{E}(||\epsilon_t||) + \sqrt{\mathbb{E}(\frac{1}{n^2}\sum_{i=1}^n\sum_{j=1}^d|\operatorname{Sign}(\nabla f(x_t^j)) - \frac{\gamma_1}{KG}v_t^{i,j}|^2)} \\
&\leq \mathbb{E}(||\epsilon_t||) + 2\sqrt{\frac{d}{n}}.
\end{aligned}
\tag{8}
$$

Taking the expectation of Eq. (8), we have

$$\mathbb{E}(f(x_{t+1})) - \mathbb{E}(f(x_t)) \leq -\gamma_1\mathbb{E}(||\nabla f(x_t)||_1) + 3G\gamma_1\sqrt{d}\mathbb{E}(||\epsilon_t||) + 6G\gamma_1\frac{d}{\sqrt{n}} + 2L\gamma_1^2 d. \tag{9}$$

Decomposing $\epsilon_t$, we have

$$
\begin{aligned}
\epsilon_t &= \frac{\gamma_1\eta}{G}\nabla f(x_t) + \frac{\gamma_1}{nKG}\sum_{i=1}^{n}v_t^i \\
&= \frac{\gamma_1\eta}{Gn}\sum_{i=1}^{n}\nabla F_i(x_t) + \frac{\gamma_1}{nKG}\sum_{i=1}^{n}v_t^i \\
&= \frac{1}{n}\sum_{i=1}^{n}(\frac{\gamma_1\eta}{G}\nabla F_i(x_t) + \frac{\gamma_1}{KG}v_t^i) \\
&= \frac{1}{n}\sum_{i=1}^{n}\epsilon_t^i.
\end{aligned}
\tag{10}
$$

We further define $h_t^i = -\frac{1}{KG}\sum_{k=0}^{K-1}\nabla F_i(y_{t,k}^i; \xi_{t,k}^i)$, $\delta_t^i = h_t^i + \frac{1}{G}\nabla F_i(x_t)$.

Referring to Algorithm 1, we have $v_t^i = \beta_2 v_{t-\tau^i}^i + (\beta_1 - \beta_2)g_{t-\tau^i}^i + (1 - \beta_1)g_t^i$.

For each client $i$, we have

$$
\begin{aligned}
\frac{\gamma_1}{KG}v_t^i &= \frac{\gamma_1}{KG}\beta_2 v_{t-\tau^i}^i + \gamma_1\eta(\beta_1 - \beta_2)h_{t-\tau^i}^i + \gamma_1\eta(1 - \beta_1)h_t^i \\
&= \beta_2(\epsilon_{t-\tau^i}^i - \frac{\gamma_1\eta}{G}\nabla F_i(x_{t-\tau^i})) + \gamma_1\eta(\beta_1 - \beta_2)(\delta_{t-\tau^i}^i - \frac{1}{G}\nabla F_i(x_{t-\tau^i})) \\
&\quad + \gamma_1\eta(1 - \beta_1)(\delta_t^i - \frac{1}{G}\nabla F_i(x_t)).
\end{aligned}
\tag{11}
$$

Converting the form of Eq. (11), we have

$$\epsilon_t^i = \beta_2\epsilon_{t-\tau^i}^i + \gamma_1\eta(\beta_1 - \beta_2)\delta_{t-\tau^i}^i + \gamma_1\eta(1 - \beta_1)\delta_t^i + \underbrace{\frac{\gamma_1\eta\beta_1}{G}\nabla F_i(x_t) - \frac{\gamma_1\eta\beta_1}{G}\nabla F_i(x_{t-\tau^i})}_{s_t^i}. \tag{12}$$

Taking the $\ell_2$ norm of $s_t^i$, with Assumption 4.1, we have

$$
\begin{aligned}
||s_t^i|| &= \frac{\gamma_1\eta\beta_1}{G}||\nabla F_i(x_t) - \nabla F_i(x_{t-\tau^i})|| \\
&\leq \frac{\gamma_1\eta}{G}||\nabla F_i(x_t) - \nabla F_i(x_{t-\tau^i})|| \\
&\leq \frac{L\gamma_1\eta}{G}||x_t - x_{t-\tau^i}|| \\
&\leq \frac{2L\tau^i\gamma_1^2\eta\sqrt{d}}{G}.
\end{aligned}
\tag{13}
$$

Taking the expectation of $||\delta_t^i||^2$ and using the Lemma B.1, we have

$$
\begin{aligned}
\mathbb{E}(||\delta_t^i||^2) &= \mathbb{E}(|| - \frac{1}{KG} \sum_{k=0}^{K-1} \nabla F_i(y_{t,k}^i; \xi_{t,k}^i) + \frac{1}{G} \nabla F_i(x_t)||^2) \\
&\leq \frac{K \sum_{k=0}^{K-1} \mathbb{E}||\nabla F_i(y_{t,k}^i; \xi_{t,k}^i) - \nabla F_i(x_t)||^2}{K^2 G^2} \\
&\leq \frac{L^2 K \sum_{k=0}^{K-1} \mathbb{E}||y_{t,k}^i - x_t||^2}{K^2 G^2} \\
&\leq \frac{L^2(8K\eta^2\sigma_l^2 + 16K^2\eta^2\sigma_l^2 + 16K^2\eta^2 G^2)}{G^2}.
\end{aligned}
\tag{14}
$$

Taking the $\ell_2$ norm of Eq. (12) and using Eq. (13), let $\tau_0^i = 0, \tau_1^i = \tau^i, \sum_{j=0}^c \tau_j^i = \tau_{c^i}, \max\{\tau_j^i\}_{1 \leq j \leq c+1} = \tau_{max}^i, \tau_{c^i} > t-1, c = c^i \leq t-1$ and we have

$$
\begin{aligned}
||\epsilon_t^i|| &\leq ||\beta_2\epsilon_{t-\tau^i}^i|| + ||s_t^i|| + ||\gamma_1(\beta_1 - \beta_2)\delta_{t-\tau^i}^i|| + ||\gamma_1(1-\beta_1)\delta_t^i|| \\
&= ||\beta_2^{c^i+1}\epsilon_0^i|| + ||\sum_{j=0}^{c^i} \beta_2^j s_{t-\tau_j}^i|| + ||\gamma_1(\beta_1 - \beta_2) \sum_{j=0}^{c^i} \beta_2^j \delta_{t-\tau_{j+1}}^i|| + ||\gamma_1(1-\beta_1) \sum_{j=0}^{c^i} \beta_2^j \delta_{t-\tau_j}^i|| \\
&\leq \beta_2^{c^i+1}||\epsilon_0^i|| + \frac{2L\tau_{max}^i\gamma_1^2\eta\sqrt{d}}{G} \sum_{j=0}^{c^i} \beta_2^j + \gamma_1(\beta_2 - \beta_1)||\sum_{j=0}^{c^i} \beta_2^j \delta_{t-\tau_{j+1}}^i|| + \gamma_1(1-\beta_1)||\sum_{j=0}^{c^i} \beta_2^j \delta_{t-\tau_j}^i|| \\
&\leq \beta_2^{c^i+1}||\epsilon_0^i|| + \frac{2L\tau_{max}^i\gamma_1^2\eta\sqrt{d}}{G(1-\beta_2)} + \gamma_1(\beta_2 - \beta_1)||\sum_{j=0}^{c^i} \beta_2^j \delta_{t-\tau_{j+1}}^i|| + \gamma_1(1-\beta_1)||\sum_{j=0}^{c^i} \beta_2^j \delta_{t-\tau_j}^i||.
\end{aligned}
\tag{15}
$$

Notice that the random variables $(\delta_t^i)_{1 \leq t \leq T}$ are independent, so $\mathbb{E}\langle \delta_{t1}^i, \delta_{t2}^i \rangle = 0$. Taking the expectation of $||\sum_{j=0}^{c^i} \beta_2^j \delta_{t-\tau_j}^i||, ||\sum_{j=0}^{c^i} \beta_2^j \delta_{t-\tau_{j+1}}^i||$ and using Eq. (14), we have

$$
\begin{aligned}
\mathbb{E}||\sum_{j=0}^{c^i} \beta_2^j \delta_{t-\tau_j}^i|| = \mathbb{E}||\sum_{j=0}^{c^i} \beta_2^j \delta_{t-\tau_{j+1}}^i|| &\leq \sqrt{\mathbb{E}(||\sum_{j=0}^{c^i} \beta_2^j \delta_{t-\tau_j}^i||^2)} \\
&= \sqrt{\mathbb{E}(\sum_{j=0}^{c^i} \beta_2^{2j} ||\delta_{t-\tau_j}^i||^2)} \\
&\leq \sqrt{\frac{1}{1-\beta_2} \frac{L^2(8K\eta^2\sigma_l^2 + 16K^2\eta^2\sigma_l^2 + 16K^2\eta^2 G^2)}{G^2}}.
\end{aligned}
\tag{16}
$$

Taking the expectation of Eq. (15) and substituting Eq. (16) in it, we further have

$$
\mathbb{E}||\epsilon_t^i|| \leq \beta_2^{c^i+1}||\epsilon_0^i|| + \frac{2L\tau_{max}^i\gamma_1^2\eta\sqrt{d}}{G(1-\beta_2)} + 2\gamma_1(1-\beta_1)\sqrt{\frac{1}{1-\beta_2} \frac{L^2(8K\eta^2\sigma_l^2 + 16K^2\eta^2\sigma_l^2 + 16K^2\eta^2 G^2)}{G^2}}.
\tag{17}
$$

Recursively iterating it from $t = 0$ to $t = T$ and substituting Eq. (17) into Eq. (10), we have

$$
\begin{aligned}
\frac{1}{T} \sum_{t=1}^{T} \mathbb{E}(\|\nabla f(x_t)\|_1) &\leq \frac{f(x_0) - \min f}{\gamma_1 T} + 3G\sqrt{d}\mathbb{E}(\|\epsilon_t\|) + 6G\frac{d}{\sqrt{n}} + 2L\gamma_1 d \\
&\leq \frac{f(x_0) - \min f}{\gamma_1 T} + 3G\sqrt{d}\frac{\sum_{t=1}^{T}\sum_{i=1}^{n}\mathbb{E}\|\epsilon_t^i\|}{nT} + 6G\frac{d}{\sqrt{n}} + 2L\gamma_1 d \\
&\leq \frac{f(x_0) - \min f}{\gamma_1 T} + 3G\sqrt{d}\frac{\sum_{t=1}^{T}\sum_{i=1}^{n}\beta_2^{c^i+1}\|\epsilon_0^i\|}{nT} + \frac{6L\tau_{max}\gamma_1^2\eta d}{1-\beta_2} \\
&\quad + 6G\sqrt{d}\gamma_1(1-\beta_1)\sqrt{\frac{1}{1-\beta_2}\frac{L^2(8K\eta^2\sigma_l^2 + 16K^2\eta^2\sigma_l^2 + 16K^2\eta^2 G^2)}{G^2}} \\
&\quad + 6G\frac{d}{\sqrt{n}} + 2L\gamma_1 d \\
&\leq \frac{f(x_0) - \min f}{\gamma_1 T} + \frac{3G\sqrt{d}\phi}{nT(1-\beta_2)} + \frac{6L\tau_{max}\gamma_1^2\eta d}{1-\beta_2} \\
&\quad + 12L\gamma_1\eta(1-\beta_1)\sqrt{\frac{d}{1-\beta_2}(2K\sigma_l^2 + 4K^2\sigma_l^2 + 4K^2 G^2)} \\
&\quad + 6G\frac{d}{\sqrt{n}} + 2L\gamma_1 d,
\end{aligned}
\tag{18}
$$

where $\tau_{max} = \max\{\tau_{max}^i\}_{1\leq i\leq m}$, $\phi = \sum_{i=1}^{m}\|\epsilon_0^i\|$ when $1 \leq t \leq T$.

Finally, when $\gamma_1 = \frac{1}{L\sqrt{T}}$ and $1 - \beta_1 = \frac{1}{\sqrt{T}}$, we complete the proof that

$$
\begin{aligned}
\frac{1}{T} \sum_{t=1}^{T} \mathbb{E}(\|\nabla f(x_t)\|_1) &\leq \frac{L(f(x_0) - \min f)}{\sqrt{T}} + \frac{3G\sqrt{d}\phi}{nT(1-\beta_2)} + \frac{6\eta d\tau_{max}}{LT(1-\beta_2)} \\
&\quad + \frac{12\eta}{T}\sqrt{\frac{d(2K\sigma_l^2 + 4K^2\sigma_l^2 + 4K^2 G^2)}{1-\beta_2}} \\
&\quad + \frac{6Gd}{\sqrt{n}} + \frac{2d}{\sqrt{T}}.
\end{aligned}
\tag{19}
$$

$\square$

**Lemma B.1.** *Let Assumption 4.1, Assumption 4.2 and Assumption 4.3 hold for $\xi_t^i$ and $\nabla F_i(\cdot;\cdot)$. Assume node i performs local SGD as*

$$
y_{t,k}^i = y_{t,k-1}^i - \eta\nabla F_i(y_{t,k-1}^i;\xi_{t,k-1}^i),
$$

*with $y_{t,0}^i = x_t$. Like the lemma proved in (Sun et al., 2023), since $0 < \eta \leq \frac{1}{4LK}$, it holds*

$$
\mathbb{E}\left\|y_{t,k}^i - x_t\right\|^2 \leq 8K\eta^2\sigma_l^2 + 16K^2\eta^2\sigma_l^2 + 16K^2\eta^2 G^2.
$$

*Proof.* Following the proof in (Sun et al., 2023), note that for any $k \in \{1,\ldots,K\}$, in client $i$,

$$
\begin{aligned}
\mathbb{E}\left\|y_{t,k}^i - x_t\right\|^2 &= \mathbb{E}\left\|y_{t,k-1}^i - \eta\nabla F_i(y_{t,k-1}^i,\xi_{t,k-1}^i) - x_t\right\|^2 \\
&\leq \mathbb{E}\|y_{t,k-1}^i - x_t - \eta\big(\nabla F_i\left(y_{t,k-1}^i;\xi_{t,k-1}^i\right) \\
&\quad - \nabla F_i\left(y_{t,k-1}^i\right) + \nabla F_i\left(y_{t,k-1}^i\right) \\
&\quad - \nabla F_i\left(x_t\right) + \nabla F_i\left(x_t\right)\big)\|^2.
\end{aligned}
\tag{20}
$$

By using the Cauchy's inequality, we have

$$\mathbb{E}\|\mathbf{a} + \mathbf{b}\|^2 \leq \left(1 + \frac{1}{\psi}\right)\mathbb{E}\|\mathbf{a}\|^2 + (1 + \psi)\mathbb{E}\|\mathbf{b}\|^2,$$

with $a = y_{t,k-1}^i - x_t - \eta\left(\nabla F_i\left(y_{t,k-1}^i; \xi_{t,k-1}^i\right) - \nabla F_i\left(y_{t,k-1}^i\right)\right)$, $b = \eta\left(\nabla F_i\left(y_{t,k-1}^i\right) - \nabla F_i(x_t) + \nabla F_i(x_t)\right)$ and $\psi = 2K - 1$.

We denote $\Re := \left(1 + \frac{1}{2K-1}\right)\mathbb{E}\|y_{t,k-1}^i - x_t - \eta\left(\nabla F_i\left(y_{t,k-1}^i; \xi_{t,k-1}^i\right) - \nabla F_i\left(y_{t,k-1}^i\right)\right)\|^2$, and $\Im := 2K\eta^2\mathbb{E}\|\nabla F_i\left(y_{t,k-1}^i\right) - \nabla F_i(x_t) + \nabla F_i(x_t)\|^2$. The unbiased expectation property of $\nabla F_i\left(y_{t,k-1}^i; \xi_{t,k}^i\right)$ gives us

$$\begin{aligned}
\Re &= \left(1 + \frac{1}{2K-1}\right)\left(\mathbb{E}\left\|y_{t,k-1}^i - x_t\right\|^2 + \eta^2\mathbb{E}\left\|\nabla F_i\left(y_{t,k-1}^i; \xi_{t,k-1}^i\right) - \nabla F_i\left(y_{t,k-1}^i\right)\right\|^2\right) \\
&\leq \left(1 + \frac{1}{2K-1}\right)\left(\mathbb{E}\left\|y_{t,k-1}^i - x_t\right\|^2 + \eta^2\sigma_l^2\right).
\end{aligned}$$

On the other hand, we have the following bound

$$\begin{aligned}
\Im &\leq 4K\eta^2\mathbb{E}\left\|\nabla F_i\left(y_{t,k-1}^i\right) - \nabla F_i(x_t)\right\|^2 + 4K\eta^2\mathbb{E}\left\|\nabla F_i(x_t)\right\|^2 \\
&\leq 4L^2K\eta^2\mathbb{E}\left\|y_{t,k-1}^i - x_t\right\|^2 + 4K\eta^2G^2.
\end{aligned}$$

When $0 < \eta \leq \frac{1}{4LK}$,

$$1 + \frac{1}{2K-1} + 4L^2K\eta^2 \leq 1 + \frac{1}{K-1},$$

and we can obtain

$$\begin{aligned}
&\mathbb{E}\left\|y_{t,k}^i - x_t\right\|^2 \\
&\leq \left(1 + \frac{1}{2K-1} + 4L^2K\eta^2\right)\mathbb{E}\left\|y_{t,k-1}^i - x_t\right\|^2 + 2\eta^2\sigma_l^2 + 4K\eta^2\sigma_l^2 + 4K\eta^2G^2 \\
&\leq \left(1 + \frac{1}{K-1}\right)\mathbb{E}\left\|y_{t,k-1}^i - x_t\right\|^2 + 2\eta^2\sigma_l^2 + 4K\eta^2\sigma_l^2 + 4K\eta^2G^2.
\end{aligned}$$

The recursion from $j = 0$ to $K$ yields

$$\begin{aligned}
\mathbb{E}\left\|y_{t,k}^i - x_t\right\|^2 &\leq \sum_{j=0}^{K-1}\left(1 + \frac{1}{K-1}\right)^j\left[2\eta^2\sigma_l^2 + 4K\eta^2\sigma_l^2 + 4K\eta^2G^2\right] \\
&\leq (K-1)\left[\left(1 + \frac{1}{K-1}\right)^K - 1\right] \times \left[2\eta^2\sigma_l^2 + 4K\eta^2\sigma_l^2 + 4K\eta^2G^2\right] \\
&\leq 8K\eta^2\sigma_l^2 + 16K^2\eta^2\sigma_l^2 + 16K^2\eta^2G^2,
\end{aligned}$$

where we used the inequality $\left(1 + \frac{1}{K-1}\right)^K \leq 5$ holds for any $K \geq 1$. $\qquad\square$

*Table 7.* Generalization performance comparison under various datasets with the LeNet model. Each table entry gives the average test accuracy on different training accuracy levels. "/" means it cannot reach the training accuracy. Bold numbers indicate the best performance.

| | | Top-1 Test Accuracy (%). | | | | | | | |
|---|---|---|---|---|---|---|---|---|---|
| Dataset | Training Accuracy | FedAvg | SCAFFOLD | SCALLION | FedEF-HS | FedEF-TopK | FedEF-Sign | FedLion | FedSMU |
| CIFAR-10 | 83-84 | 75.14 | **77.49** | 75.9 | 75.63 | 75.48 | 75.7 | 77.22 | 77.1 |
| | 85-86 | 76.17 | 76.37 | 76.87 | 76.83 | 76.83 | 77.4 | 78.3 | **78.39** |
| | 87-88 | 78.56 | 79.24 | 77.79 | 77.85 | 77.74 | / | 79.23 | **79.78** |
| CIFAR-100 | 66-67 | 40.08 | 45.76 | 40.56 | / | 41.28 | / | 45.55 | **49.03** |
| | 68-69 | 40.6 | 46.16 | 40.88 | / | 41.81 | / | 46.15 | **50.04** |
| | 70-71 | 40.9 | 46.76 | 41.23 | / | 42.34 | / | 46.56 | **50.85** |

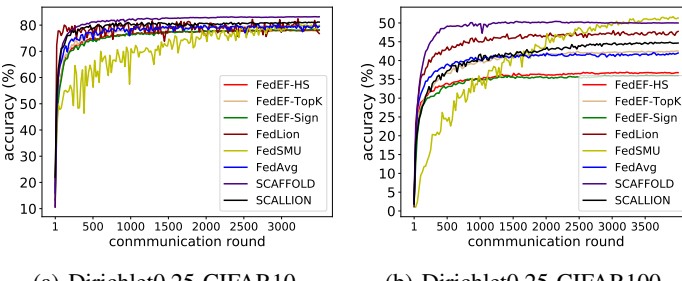

(a) Dirichlet0.25-CIFAR10      (b) Dirichlet0.25-CIFAR100

*Figure 3.* Convergence performance vs. number of communication rounds on CIFAR-10 and CIFAR-100, with 100 clients and 10% participation, using LeNet model for different algorithms.

## C. Additional Experiments

### C.1. Measure of Generalization

In the above experimental results in the main text, generalization refers to an algorithm's ability to achieve top test accuracy, where the test dataset is different from the training dataset. Furthermore, we consider an additional perspective on generalization to further evaluate the performance of our FedSMU algorithm. Here, generalization refers to a model's ability to achieve test performance at similar training error levels. Based on this definition, we compare the validation performance at similar training accuracy levels. The results in Table 7 show that on the CIFAR-10 and CIFAR-100 datasets with LeNet model, the FedSMU algorithm achieves the highest test accuracy and demonstrates the best generalization performance.

### C.2. Convergence Performance vs. Communication Rounds

In Section 5.2.1, considering that FedSMU is a compression algorithm, we compare the convergence of different algorithms in terms of communication bits. However, communication rounds are also important, thus in Figure 3, we show the convergence performance in terms of communication rounds. Also, we compare the convergence rate using the number of communication rounds required to achieve the target accuracy and the results are presented in the Table 8.

For Shakespeare, our algorithm does not require more communication rounds compared to most algorithms. However, for CIFAR-10 and CIFAR-100, it slightly exceeds the number of rounds needed by other algorithms. This may be because the distribution of image data is more complex, with each sample containing a large amount of pixel information. Training with such highly heterogeneous data results in the gradients that, after taking the sign, introduce noises in the training process, thereby slowing down the convergence. However, from a long-term perspective, these noises can lead to an improved model performance.

### C.3. Convergence Performance vs. Wall-Clock Time

We test the wall-clock time needed for each baseline to execute one communication round. Taking CIFAR-100 and participation rate $\frac{n}{m} = 0.1$ as an example, the average wall-clock time required to execute a round is as follows: FedSMU (10.43 seconds), FedAvg (10.15 seconds), FedEF-HS (10.46 seconds), FedLion (10.63 seconds), SCAFFOLD (10.38 seconds). Experiments demonstrate that in a single communication round, our algorithm introduces no significantly

*Table 8.* Number of communication rounds to achieve a preset target accuracy with 100 clients and 10% participation. CIFAR-10 and CIFAR-100 use the LeNet model and Shakespeare uses the RNN network. "/" means it cannot reach the training accuracy. Bold numbers indicate the best performance.

| Dataset | Training Accuracy (%) | FedAvg | SCAFFOLD | SCALLION | FedEF-HS | FedEF-TopK | FedEF-Sign | FedLion | FedSMU |
|---|---|---|---|---|---|---|---|---|---|
| CIFAR-10 (Dir0.25) | 55 | 26 | 29 | 23 | 41 | 37 | 50 | **21** | 65 |
| | 60 | 43 | 43 | **29** | 62 | 57 | 81 | 33 | 118 |
| | 65 | 57 | 62 | 48 | 108 | 96 | 111 | **50** | 288 |
| CIFAR-100 (Dir0.25) | 35 | 193 | **86** | 286 | 690 | 355 | 794 | 100 | 832 |
| | 40 | 629 | **142** | 730 | / | 882 | / | 225 | 1218 |
| | 45 | / | **270** | 3703 | / | / | / | 632 | 1811 |
| Shakespeare (noniid) | 25 | 17 | 12 | 13 | 30 | 20 | 57 | **10** | 11 |
| | 30 | 32 | 19 | 20 | 45 | 36 | 78 | **17** | 20 |
| | 35 | 61 | **27** | **27** | 85 | 68 | 177 | 28 | 48 |

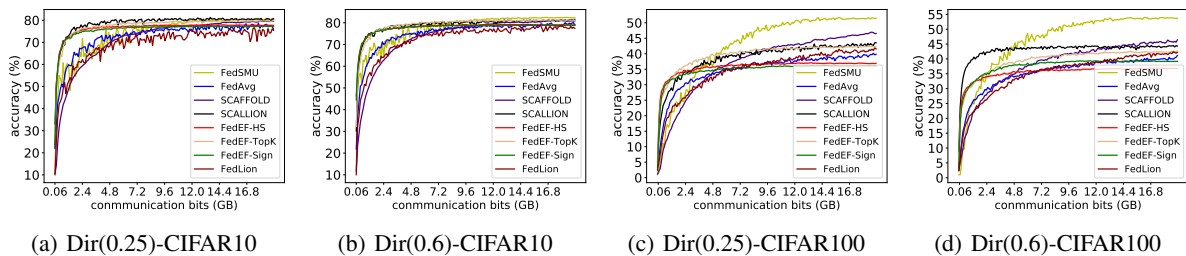

(a) Dir(0.25)-CIFAR10    (b) Dir(0.6)-CIFAR10    (c) Dir(0.25)-CIFAR100    (d) Dir(0.6)-CIFAR100

*Figure 4.* Convergence performance vs. number of communication rounds on CIFAR-10, CIFAR-100 dataset and LeNet model, with 100 clients and 10% participation.

additional time overhead compared to other algorithms. Therefore, the results using wall-clock time as a metric are similar to those measured by communication rounds. We will not include a separate plot here and please refer to Figure 3 and Table 8.

## C.4. Convergence Performance Considering Uplink Communications

In Figure 2, we only consider uplink (client-to-server) communication cost and assume the downlink communication overhead to be the same. Here, we can define the total communication cost per round as presented in (Condat et al., 2022):

$$\text{Total Communication} = \text{Uplink Communication} + c \cdot \text{Downlink Communication}, c \in [0, 1].$$

In practice, due to the factors such as system asymmetry, caching constraints and protocol limitations, the uplink speed is often significantly lower than the downlink speed, as discussed in (Condat et al., 2023). Consequently, many communication-efficient FL studies (Li & Li, 2023; Richtárik et al., 2021) focus merely on minimizing the uplink cost alone.

To have a more comprehensive evaluation of our FedSMU, we followed the setting in (Condat et al., 2023) and set $c = 0.1$, and depicted Figure 4 to compare the total communication cost including both the upload and download bits. It shows that FedSMU remains communication-efficient even when accounting for the downlink overhead, with a significantly lower total cost compared to the other baselines at comparable accuracy levels.

## C.5. Discussion on $\alpha$-bit

Here, we further discuss the extension from the 1-bit compression to an $\alpha$-bit one. First, we would like to acknowledge that for the general quantization-based compression algorithms, a higher precision quantization may often lead to a faster convergence. However, this may not hold for our FedSMU. This conclusion is based on our analysis, as follows.

1) Both experimentally (Figure 1) and intuitively, the symbolic operation (i.e., 1-bit quantization) helps alleviate the heterogeneity of model updates, as all updates have uniform magnitude across all dimensions for each client. Furthermore, reducing model heterogeneity should also intuitively contribute to improving model performance in heterogeneous federated settings.

*Table 9.* Number of communication rounds to achieve a preset target test accuracy with Dirichlet-0.25 on CIFAR-10 dataset with LeNet model. "/" means it cannot reach the test accuracy and bold numbers indicate the smallest rounds.

| Number of rounds needed for achieving a target test accuracy. | | | |
|---|---|---|---|
| Test Accuracy(%) | 1-bit (FedSMU) | 3-bit | 8-bit |
| 40 | **26** | 65 | 54 |
| 45 | **33** | 91 | 68 |
| 50 | **50** | 168 | 119 |
| 55 | **65** | 285 | 185 |
| 60 | **118** | 456 | 224 |
| 65 | **288** | 833 | 342 |
| 67.5 | **399** | 1260 | 401 |
| 69 | 507 | 1967 | **456** |
| 72.3 | **642** | / | 643 |
| 75 | 1142 | / | **1040** |
| 77.5 | **1746** | / | 1979 |

*Table 10.* Number of communication rounds to achieve a preset target test accuracy with Dirichlet-0.25 on CIFAR-10 dataset and different CNN model. $d_1$ and $d_2$ indicate small and large dimension while L and H indicate low and high participation rates. Bold numbers indicate the smallest rounds.

| Number of rounds needed for achieving a target test accuracy. | | | |
|---|---|---|---|
| Test Accuracy(%) | $d_1$, H | $d_2$, H | $d_1$, L |
| 40 | **26** | 30 | 46 |
| 45 | **33** | 35 | 112 |
| 50 | 50 | **43** | 193 |
| 55 | **65** | 65 | 226 |
| 60 | 118 | **111** | 384 |
| 65 | 288 | **275** | 899 |
| 67.5 | **399** | 409 | 949 |
| 69 | **507** | 507 | 1025 |
| 72.3 | **642** | 769 | 1752 |
| 75 | **1142** | 1185 | 2221 |
| 77.5 | 1746 | **1745** | 2878 |

2) However, such a sign operation (e.g., signSGD) alone does not directly improve generalization in experiments. Inspired by Lion optimizer (Chen et al., 2024) that incorporates the sign operation and then enhances the convergence and generalization to learn in central learning, we introduce Lion's structure into federated learning and verify that this combination can indeed improve model generalization.

Consequently, we conclude that in our optimized structure, 1-bit quantization outperforms higher-bit quantization, since multi-bit compression does not guarantee that the update amplitude of each client is consistent. Experimental results in Table 9 further validate that for our designed optimization algorithm, using a higher-bit compression may not enhance the algorithm's convergence or generalization.

### C.6. Factors Influencing Convergence Speed

Theoretically, a lower client participation rate (i.e., a larger $\tau_{max}$) leads to a slower algorithm convergence. Similarly, a higher model dimension $d$ also results in a slower algorithm convergence. To validate this, we have conducted the following experiments.

All of these experiments are done on CIFAR-10 dataset with Dirichlet-0.25. To illustrate the relationship between the model dimension and convergence rate, we use two different CNN models ($d_1 = 797248$ and $d_2 = 1723648$) to study the impact of model dimension. Note that here we modify the size of the convolutional layers, keeping the model depth constant. The

*Table 11.* Top accuracy (%) comparison between ablation experiments on CIFAR-100 dataset (Dirichlet-0.25) with LeNet model, where NTC indicates the number of total clients, and PR indicates the participation rate.

| NTC / PR | FedSMU | FedSMUMC |
|----------|--------|----------|
| 100 / 0.1 | 52.35 | **52.53** |
| 100 / 0.03 | 51.87 | **52.14** |

*Table 12.* Top validation accuracy (%) on CIFAR-100 dataset with Dirichlet 0.25 and LeNet model, with 100 clients and 10% participation rate.

| | | | Top-1 Test Accuracy (%). | | | | |
|---------|---------|--------|--------------|---------------|----------------------|----------------------|----------------------|
| Dataset | Setting | FedSMU | Fed-LocalLion | Fed-GlobalLion | FedSMU($\gamma_2 = 0$) | FedSMU($\beta_1 = 0$) | FedSMU(full-precision) |
| CIFAR-100 | Dir (0.25) | **52.35** | 36.77 | 47.94 | 51.34 | 28.03 | 42.67 |

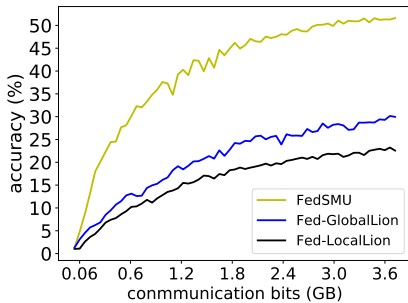

*Figure 5.* Convergence performance vs. number of communication bits on CIFAR-100 dataset and LeNet model, with 100 clients and 10% participation rates, Dirichlet-0.25 for different ablation algorithms and FedSMU.

number of clients is 100 with the participation ratio of 0.1. To illustrate the relationship between participation rate and convergence rate, We use the CNN model with $d_1 = 797248$ and set the number of clients as 100 with different participation ratio ($\frac{n}{m} = 0.03$ and $\frac{n}{m} = 0.1$, represented in the Table 10 by L and H) to demonstrate the influence of client participation rate.

The experimental results in Table 10 show that when the participation rate is higher (i.e., the $\tau_{max}$ is smaller) and the dimension is smaller, the convergence speed can be faster, which matches with the Theorem 4.4 and is intuitional.

### C.7. Variants Solving Momentum Staleness

While our convergence analysis and experimental results demonstrate that FedSMU's performance is less affected by the client participation rate, the momentum of clients may still be extremely stale due to the partial participation in FL. In light of this, we design a variant, named FedSMUMC, to examine the impact of this momentum staleness on the generalization performance. For FedSMUMC, clients upload 1-bit model updates along with extra momentum in the full precision. The server then aggregates that momentum to update the global momentum and broadcasts it at the next round as the initial momentum for the participating clients. See Appendix D for the detail of this algorithm. Results in Table 11 indicate that by appropriately completing the momentum, we can marginally enhance the model performance, but it necessitates additional transmission of momentum with the full precision. Consequently in this sense, the local momentum staleness has a minimum impact on the global model's performance.

### C.8. Ablation Algorithms

To verify the effectiveness of different FL algorithms built upon the Lion optimizer in terms of the generalization and compression performance, we design additional variants of FL incorporated with Lion, namely Fed-LocalLion and Fed-GlobalLion. Specifically, Fed-LocalLion executes the Lion optimizer locally in parallel at clients, with the server performing model aggregation via a weighted summation. On the other hand, Fed-GlobalLion conducts the vanilla SGD locally, treats model aggregation as a pseudo-gradient on the server side, and updates the global model through the Lion optimizer. See

Appendix D for the detail of these two algorithms.

Our FedSMU consistently outperforms the other variants, as illustrated in Figure 5 and Table 12. It is worth noting that FedSMU also integrates additional model compression, whereas these variants require the same communication overhead as FedAvg. This suggests that our FedSMU design effectively harnesses the benefits of Lion, enhancing the generalization while compressing the communication load.

To further assess the necessity of key components in FedSMU, we conduct a systematic ablation study by removing specific elements and comparing each pruned variant against the full algorithm. In particular, we examine the effect of excluding the following components: 1) server-side weight decay regularization ($\gamma_2$), 2) client-side gradient sliding average ($\beta_1$), and 3) client-side gradient symbolization.

The experimental results in Table 12 demonstrate that the client-side sliding average plays the most crucial role in achieving a stable and effective training. Additionally, the model update symbolization mechanism itself proves to be more effective than using the full-precision updates. This validates our motivation of proposing FedSMU that symbolization balances contributions of the heterogeneous clients by suppressing some extreme update magnitudes, which thus enhances the aggregation stability and leads to a better generalization.

### C.9. Comparison with Distributed Lion (Liu et al., 2024)

Here, we clarify the differences and advantages of our FedSMU compared to the D-Lion (Liu et al., 2024), as follows.

**1) Motivation.** Our FedSMU can simultaneously mitigate data heterogeneity and reduce communication compression through the symbolic operations. The analysis was carried out and verified by experiments (Figure 1). While D-Lion only considers to compress the communication.

**2) Scope of application.** Our FedSMU can deal with scenarios involving the partial client participation and multiple local updates, whereas D-Lion can not. Performing multiple local updates, in the federated settings, can effectively reduce the communication frequency and thus the overall traffic. Experimental results in Table 13 and Table 14 demonstrate that D-Lion fails in such scenarios with low client participation rates and multiple local updates, whereas FedSMU remains robust and performs well under these conditions.

**3) Algorithms design.** While both algorithms are based on the Lion optimizer, FedSMU fully leverages the structural advantages of the Lion optimizer, including weight decay in the global aggregation. In contrast, D-Lion primarily incorporates the momentum sliding averaging and symbolic operations at local update. This comprehensive utilization of the Lion optimizer structure may explain why the experimental performance of our FedSMU surpasses that of D-Lion.

**4) Compatibility with majority vote.** We have further extended FedSMU with majority vote, as FedSMU-MV. Experimental results show that FedSMU-MV achieves an accuracy of 47.66% on CIFAR-100, slightly lower than FedSMU's 51.79% under the same settings (number of clients = 100, participation rate = 0.1, Dirichlet = 0.25). This indicates that majority vote is compatible with our algorithm. The slight accuracy drop may result from FedSMU's symbolic model updates. Applying majority vote to the 1-bit results could further suppress some clients' model update information due to the dominant update direction.

Below, we provide the details of the hyperparameters used in our experiments.

- To ensure a fair comparison, both algorithms are evaluated on the CIFAR-10 and CIFAR-100 datasets, using non-IID data (Dirichlet distribution with a parameter of 0.25), with a total of 10 clients anda batch size of 50.

- For FedSMU and FedAvg, we adopt the same parameter settings as outlined in Appendix A.

- For D-Lion, we performed a grid search. The learning rate ($\epsilon$) is selected from $\{0.00005, 0.0005, 0.005, 0.015\}$ , the weight decay ($\lambda$) is chosen from $\{0.0005, 0.005, 0.001, 0.01\}$ and $\beta_1$ $\beta_2$ are selected from $\{0.9, 0.99\}$. For Table 13, the selected values are $\epsilon = 0.0005$, $\lambda = 0.001$, $\beta_1 = 0.9$, $\beta_2 = 0.99$. For Table 14, the selected values are $\epsilon = 0.015$, $\lambda = 0.01$, $\beta_1 = 0.9$, $\beta_2 = 0.9$.

Specifically, from the result in Table 13 and Table 14, we have following observations.

- With full participation and one local update (i.e., $K = 1$ with F), FedSMU performs slightly worse than D-Lion. However, in scenarios with a partial participation, FedSMU consistently outperforms D-Lion. This is intuitive, as

*Table 13.* Performance comparison on CIFAR-10 and CIFAR-100 datasets with LeNet model, where F and P indicate full and partial participation rates, and K is the number of local updates.

| Top-1 Test Accuracy (%) . | | | | | | |
|---|---|---|---|---|---|---|
| Dataset | Setting | Algorithm | $K = 1$ with F | $K = 1$ with P | $K = 5$ with F | $K = 5$ with P |
| CIFAR-10 | Dir-0.25 | FedSMU | 32.47 | 38.35 | **77.97** | **75.14** |
| | | D-Lion | 34.03 | 24.58 | 77.62 | 34.48 |
| | | FedAvg | **79.64** | **74.99** | 72.02 | 71.48 |
| | iid | FedSMU | 81.84 | **77.99** | 82.37 | **81.71** |
| | | D-Lion | **82** | 29.06 | 82.36 | 44.05 |
| | | FedAvg | 79.53 | 76.7 | 76.3 | 75.84 |
| CIFAR-100 | Dir-0.25 | FedSMU | 14.85 | 20.41 | **45.69** | **42.06** |
| | | D-Lion | 15.42 | 3.9 | 45.54 | 8.03 |
| | | FedAvg | **44.99** | **39.34** | 36.55 | 36.51 |
| | iid | FedSMU | 50.98 | **47.18** | 49.76 | **49.72** |
| | | D-Lion | **51.46** | 5.13 | **50.11** | 13.07 |
| | | FedAvg | 44.85 | 41.07 | 41.38 | 38.25 |

*Table 14.* Performance comparison on CIFAR-10 and CIFAR-100 datasets with LeNet model, where F and P indicate full and partial participation rates, and K is the number of local updates.

| Top-1 Test Accuracy (%) . | | | | | | |
|---|---|---|---|---|---|---|
| Dataset | Setting | Algorithm | $K = 100$ with F | $K = 100$ with P | $K = 500$ with F | $K = 500$ with P |
| CIFAR-10 | Dir-0.25 | FedSMU | **82.24** | **82.32** | **82.0** | **82.08** |
| | | D-Lion | 82.19 | 25.86 | 81.6 | 51.23 |
| CIFAR-100 | Dir-0.25 | FedSMU | **50.15** | **50.62** | **46.66** | **48.2** |
| | | D-Lion | 49.84 | 4.05 | 46.55 | 16.21 |

D-Lion does not maintain a complete global model at the server and only aggregates the global model updates. Thus in the partial participation settings, asynchronous clients can only save a stale global model. As a result, these clients may receive the global model updates, which, however, cannot be leveraged to recover the exact global model of the current round.

- With multiple local updates (i.e., $K > 1$), FedSMU mostly outperforms D-Lion. This performance improvement can be attributed to the different approaches to weight decay. Specifically, the hyperparameter $\gamma_2$ (denoted as $\lambda$ in D-Lion) controls the weight decay (or $L_2$ penalty) coefficient. In FedSMU, the regularization is applied to the global model $x_t$, potentially mitigating overfitting and thus enhancing generalization. In contrast, D-Lion applies this regularization to the local model $x_{t-1}^i$. As a result, when the local updates occur multiple times, D-Lion's regularization primarily affects the local model, and does not directly improve the generalization capability of the global model. Consequently, when finally evaluating the generalization performance of the global model, FedSMU demonstrates a significant advantage over D-Lion.

- In heterogeneous scenarios, the performance of both FedSMU and D-Lion is poorer than that of FedAvg, especially when $K$ is small. This is an interesting and somewhat unexpected finding, which we speculate is due to the data heterogeneity. In the heterogeneous settings, each client samples a mini-batch of data for training and performs only a single time of update, followed by the application of the sign operation to the model update. Since the local update occurs only once, it introduces a substantial sampling variance and inter-client variance. The sign operation, which normalizes the magnitude of updates, may inadvertently amplify this variance between clients, leading to an unstable or even divergent global model aggregation.

*Table 15.* Performance comparison under different datasets with LeNet model, where L and H indicate low and high participation rates. Bold numbers indicate the best performance.

| Top-1 Test Accuracy (%). | | | |
|---|---|---|---|
| Dataset | Setting | FedSMU | EF21 |
| CIFAR-10 | Dir(0.25)-L | **80.12** | 74.43 |
| | Dir(0.25)-H | 80.74 | **81.51** |
| CIFAR-100 | Dir(0.25)-H | **52.35** | 50.07 |

*Table 16.* Performance comparison under different datasets with 100 clients and 10% participation rates, Dirichlet-0.25, LeNet model. Bold numbers indicate the best performance.

| Top-1 Test Accuracy (%) . | | | |
|---|---|---|---|
| Dataset | FedSMU | FedAMS | FedCAMS |
| CIFAR-10 | 80.74 | **82.47** | 80.15 |
| CIFAR-100 | **52.35** | 47.97 | 48.3 |

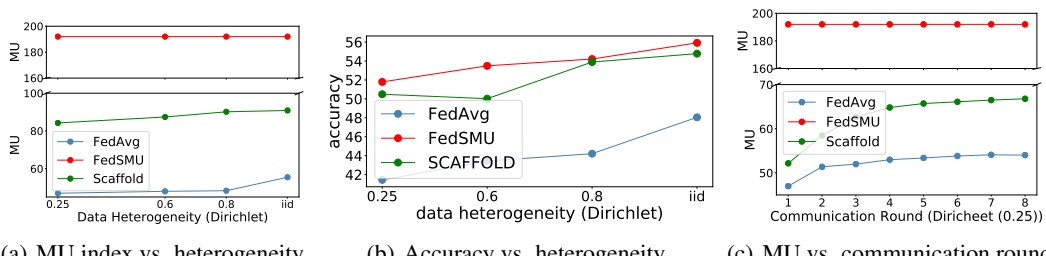

(a) MU index vs. heterogeneity     (b) Accuracy vs. heterogeneity     (c) MU vs. communication round

*Figure 6.* Magnitude uniformity (MU) index and top validation accuracy of FedAvg, SCAFFOLD and FedSMU (ours) on CIFAR-100 with LeNet model.

## C.10. Comparison with EF21 (Richtárik et al., 2021)

We further explore Error Feedback 2021 (Richtárik et al., 2021) algorithm as a state-of-the-art method for Top-K compression and make a comparison with it.

Experiments on CIFAR-10 and CIFAR-100 are conducted. We set a total of 100 clients with different participation rate (3% and 10%, represented in Table 15 by L and H) and use Dirichlet-0.25. The experimental results are shown in Table 15.

On CIFAR-100, FedSMU still shows a high performance. While on CIFAR-10, the accuracy of FedSMU can be higher than EF21 with a lower participation. These results strongly demonstrate the superiority of FedSMU in complex image classification tasks, especially under a low client participation rate, which may result from the sign operation promoting the fair contribution of clients effectively to the global model update.

## C.11. Comparison with Adaptive Algorithms

We compare with two adaptive algorithms in (Wang et al., 2022): the optimization-based FedAMS and the compression-based FedCAMS. FedAMS is designed to accelerate the convergence using momentum, while FedCAMS extends FedAMS by further compressing the upload communication. Experiments are conducted on CIFAR-10 and CIFAR-100 datasets. We use a total of 100 clients with a partial participation ratio of 0.1 and employ a Dirichlet distribution with a concentration parameter of 0.25. The experimental results are presented in the Table 16.

The experimental results demonstrate that on the CIFAR-10 dataset, FedSMU also outperforms FedCAMS but is slightly inferior to FedAMS, while FedSMU exhibits a superior performance compared to FedAMS and FedCAMS on the CIFAR-100 dataset. The results strongly demonstrate the superiority of FedSMU in complex image classification tasks, even comparable to the uncompressed federated adaptive algorithm, which may result from promoting the fair contribution of clients effectively to the global model update.

## C.12. Magnitude Uniformity Index of More Algorithms

We provide Figure 6 to show the correlation between Magnitude Uniformity (MU), data heterogeneity, and accuracy of three different algorithms.

The results indicate that with FedAvg, data heterogeneity significantly amplifies the differences in the magnitude of model updates across clients, leading to unstable global aggregation and poorer generalization performance. While SCAFFOLD reduces variance to address these differences, FedSMU directly ensures consistency across all model updates through symbolic operations. Those two approaches enhances Magnitude Uniformity among clients, ultimately improving accuracy.

# D. OTHER ALGORITHMS

FedSMUMC, as a variant evaluated in the ablation study of our FedSMU, is shown in Algorithm 2. The basic procedure is equivalent to FedSMU. At each round $t \in [T]$, a subset of clients $\mathcal{N}_t \subseteq \mathcal{M}$ are active, and the server transmits its current model $x_t$ and global momentum $M_t$ to these clients. Local clients also additionally transfer $m_t^i$ back to the server (Line 13) and average them to update the momentum for the next round (Line 16).

---

**Algorithm 2** FedSMUMC

---

**Server Initialization**: $x_1, M_1$;
**for** *each round $t = 1, 2, ...T$* **do**
    sample clients $\mathcal{N}_t \subseteq \mathcal{M}$
    **for** *each client $i \in \mathcal{N}_t$ in parallel* **do**
        receive and initialize local model $y_{t,0}^i = x_t$
        receive momentum $M_t$
        **for** *each local step $k = 1, 2, \ldots, K$* **do**
            $y_{t,k}^i = y_{t,k-1}^i - \eta \nabla F_i(y_{t,k-1}^i, \xi_{t,k-1}^i)$
        **end**
        $g_t^i = y_{t,K}^i - y_{t,0}^i$
        $u_t^i = \text{Sign}(\beta_1 M_t + (1 - \beta_1) g_t^i)$
        $m_t^i = \beta_2 M_t + (1 - \beta_2) g_t^i$
        send $u_t^i, m_t^i$ to server
    **end**
    // at server:
    $M_{t+1} = \frac{1}{n} \sum_{i=1}^n m_t^i$
    $x_{t+1} = x_t + \gamma_1(\frac{1}{n} \sum_{i=1}^n u_t^i - \gamma_2 x_t)$
    broadcast $x_{t+1}, M_{t+1}$
**end**

---

Fed-LocalLion, as a variant evaluated in the ablation study of our FedSMU, is shown in Algorithm 3. At each round $t \in [T]$, a subset of clients $\mathcal{N}_t \subseteq \mathcal{M}$ are active, and the server transmits its current model $x_t$ to these clients. Each active client then performs SGD (Line 8) and uses the Lion optimizer to further update model. The server aggregates the local model difference $\Delta_t^i$ to compute $x_{t+1}$.

Fed-GlobalLion, as a variant evaluated in the ablation study of our FedSMU, is shown in Algorithm 4. At each round $t \in [T]$, a subset of clients $\mathcal{N}_t \subseteq \mathcal{M}$ are active, and the server transmits its current model $x_t$ to these clients. Each active client then updates the local model (Line 8) and sends the model difference $g_t^i$ to server. The server aggregates the $g_t^i$ as the global model difference $G_t$ (Line 14) and uses the Lion optimizer to update.

---

**Algorithm 3** Fed-LocalLion

---

**Server Initialization**: $x_1$;
**Client Initialization**: $m_0^i = 0$;
**for** *each round $t = 1, 2, ...T$* **do**
    sample clients $\mathcal{N}_t \subseteq \mathcal{M}$
    **for** *each client $i \in \mathcal{N}_t$ in parallel* **do**
        receive and initialize local model $y_{t,0}^i = x_t$
        **for** *each local step $k = 1, 2, \ldots, K$* **do**
            $y_{t,k}^i = y_{t,k-1}^i - \eta \nabla F_i(y_{t,k-1}^i, \xi_{t,k-1}^i)$

        **end**
        $g_t^i = y_{t,K}^i - y_{t,0}^i$
        $u_t^i = \text{Sign}(\beta_1 m_{t-1}^i + (1 - \beta_1)g_t^i)$
        $m_t^i = \beta_2 m_{t-1}^i + (1 - \beta_2)g_t^i$ (for $i \notin \mathcal{N}_t, m_t^i = m_{t-1}^i$)
        $y_t^i = y_{t,K}^i + \gamma_1(u_t^i - \gamma_2 y_{t,K}^i)$
        $\Delta_t^i = y_t^i - y_{t,0}^i$
        send $\Delta_t^i$ to server
    **end**
    // at server:
    $x_{t+1} = x_t + \eta_g(\frac{1}{n}\sum_{i=1}^n \Delta_t^i)$
    broadcast $x_{t+1}$
**end**

---

---

**Algorithm 4** Fed-GlobalLion

---

**Server Initialization**: $x_1$, $M_0 = 0$;
**for** *each round $t = 1, 2, ...T$* **do**
    sample clients $\mathcal{N}_t \subseteq \mathcal{M}$
    **for** *each client $i \in \mathcal{N}_t$ in parallel* **do**
        receive and initialize local model $y_{t,0}^i = x_t$
        **for** *each local step $k = 1, 2, \ldots, K$* **do**
            $y_{t,k}^i = y_{t,k-1}^i - \eta \nabla F_i(y_{t,k-1}^i, \xi_{t,k-1}^i)$

        **end**
        $g_t^i = y_{t,K}^i - y_{t,0}^i$
        send $g_t^i$ to server
    **end**
    // at server:
    $G_t = \frac{1}{n}\sum_{i=1}^n g_t^i$
    $U_t = \text{Sign}(\beta_1 M_{t-1} + (1 - \beta_1)G_t)$
    $M_t = \beta_2 M_{t-1} + (1 - \beta_2)G_t$
    $x_{t+1} = x_t + \gamma_1(U_t - \gamma_2 x_t)$
    broadcast $x_{t+1}$
**end**

---

