# OpenReview forum: "FedSMU: Communication-Efficient and Generalization-Enhanced Federated Learning through Symbolic Model Updates"
_ICML.cc/2025/Conference — ICML 2025 poster_

### Official Review · Reviewer_AyCK · 2025-03-07

**Overall Recommendation:** 4

**Summary:**

The paper considers for federated learning in the heterogeneous regime, with compression and partial participation. A new algorithm called FedSMU is proposed.

## update after rebuttal
I think the paper deserves to be accepted and I am confident that the authors will make the recommended changes to make the paper even better.

**Claims And Evidence:**

The focus in on the non-convex setting, for deep learning. The functions are supposed smooth and with bounded gradients (Assumption 4.3).

The experiments and comparisons are satisfying and show the merits of FedSMU. The improvement is not large, however, and in some cases FedSMU is even worse than competitors. For instance it is worse than EF21 and FedAMS on CIFAR-10.

There is a theoretical analysis in Section 4, which is rare for papers focused on empirical performance in deep learning settings. This deserves to be saluted.

**Essential References Not Discussed:**

You write that "Current approaches in FL often prioritize either mitigating data heterogeneity to enhance generalization or compress-
ing model updates to alleviate communication, rather than addressing both challenges concurrently."
There are however works combining local steps, with control variates to mitigate client drift due to heterogeneity, and compression.
* CompressedScaffnew in Condat et al. “Provably Doubly Accelerated Federated Learning: The First Theoretically Successful Combination of Local Training and Compressed Communication,” preprint arXiv:2210.13277, 2022.
* TAMUNA, which extends CompressedScaffnew to partial participation, in Condat et al. “TAMUNA: Doubly Accelerated Federated Learning with Local Training, Compression, and Partial Participation,” preprint arXiv:2302.09832, 2023.
* LoCoDL, which uses arbitrary unbiased compression, unlike CompressedScaffnew and TAMUNA that use specific sparsification, in Condat et al. “LoCoDL: Communication-Efficient Distributed Learning with Local Training and Compression,” ICLR 2025.

These methods achieve acceleration in the convex setting. In the non-convex setting, it is less clear how to mitigate client drift. This has been studied in
* Yi et al. “FedComLoc: Communication-Efficient Distributed Training of Sparse and Quantized Models,” preprint arXiv:2403.09904, 2024.
* Meinhardt et al. “Sparse-ProxSkip: Accelerated Sparse-to-Sparse Training in Federated Learning,” preprint arXiv:2405.20623, 2024.

There is the important paper Douillard et al. "DiLoCo: Distributed Low-Communication Training of Language Models," arXiv:2311.08105, 2023. Instead of SGD local steps, it uses AdamW for the inner iterations, with good empirical performance.

**Experimental Designs Or Analyses:**

The experiments seem valid to me.

**Methods And Evaluation Criteria:**

The evaluation is good.

**Other Comments Or Suggestions:**

No

**Other Strengths And Weaknesses:**

The paper is well written.

**Questions For Authors:**

I don't have questions. The paper is good overall.

**Relation To Broader Scientific Literature:**

The literature is correctly reviewed. I suggest below some references to add.

**Theoretical Claims:**

I did not check the details of the theoretical results but the statements make sense and seem correct.

---

> ### Author Rebuttal · Authors · 2025-04-01
>
> We would like to thank the reviewer for the comments. In the following, we have provided our detailed responses to these comments.
>
> > Experiments on CIFAR-10
>
> We thank the reviewer for this observation. Our experimental results indicate that FedSMU achieves a notably better performance on more complex datasets, such as CIFAR-100 and Tiny-ImageNet, while the improvement on CIFAR-10 is relatively marginal. We attribute this to the lower data complexity and heterogeneity in CIFAR-10, which restrains the potential benefit of FedSMU’s core designs, particularly its ability in handling client update imbalance and data heterogeneity. As shown in Table 3 of the original manuscript, FedSMU still outperforms all the baselines on Tiny-ImageNet, highlighting that its advantages become more pronounced in challenging federated scenarios.
>
> > There are however works combining local steps, with control variates to mitigate client drift due to heterogeneity, and compression.
>
> We thank the reviewer for the insightful comment and for suggesting the relevant literature. We agree that there are existing works that address both the data heterogeneity and communication efficiency, particularly in the convex setting. In response, we will revise our original statement as follows.
>
> “Some existing approaches in FL address either data heterogeneity to improve generalization or communication overhead through update compression. However, it remains a challenge to jointly addressing both of them, especially under the non-convex settings. For example, CompressedScaffnew [D1] and TAMUNA [D2] combine control variates (used to mitigate client drift due to heterogeneity) with the model compression. However, these methods rely on permutation-based compression schemes, which are relatively complex and less flexible. LoCoDL [D3] extends this line of work by supporting a broader class of compressors and demonstrating a convergence acceleration in convex problems, but it focuses exclusively on the convex setting. Our work complements these efforts by proposing a unified approach that simultaneously improves generalization and reduces communication overhead in the more challenging non-convex regime.”
>
> > In the non-convex setting, it is less clear how to mitigate client drift.
>
> In our work, we propose a new mechanism, symbolization of client updates, which normalizes the magnitude of each model parameter before transmission. This design aims to balance the contribution from each client, reducing the influence of extreme updates caused by local data heterogeneity, and thereby mitigating the model drift during aggregation. The effectiveness of this approach is empirically demonstrated in Figure 1. Moreover, relevant studies, such as FedComLoc [D4] and Sparse-ProxSkip [D5], make attempts to explore client drift under the non-convex objectives. However, FedComLoc’s performance may degrade under the compressed communication due to its reliance on communication variables, while Sparse-ProxSkip assumes the full client participation, which may not always be feasible in real-world FL scenarios.
>
> > Works on distributed low-communication language models
>
> We thank the reviewer for suggesting this work. We will incorporate DiLoCo [D6] into the Related Work section. This method adopts AdamW for local updates and Nesterov momentum globally, demonstrating a strong empirical performance. While our current focus is on symbolized updates within a communication-efficient FL framework, integrating advanced optimization techniques, such as those in [D6] (especially adaptive optimizers like AdamW), could further enhance the model performance. We consider this as a promising direction for our future work.
>
> [D1] Condat et al. "Provably doubly accelerated federated learning: The first theoretically successful combination of local training and communication compression," arXiv preprint *arXiv:2210.13277* (2022).
>
> [D2] Condat et al. "Tamuna: Doubly accelerated federated learning with local training, compression, and partial participation," *International Workshop on Federated Learning in the Age of Foundation Models in Conjunction with NeurIPS*, 2023.
>
> [D3] Condat et al. "Locodl: Communication-efficient distributed learning with local training and compression," ICLR 2025.
>
> [D4] Yi et al. “FedComLoc: Communication-Efficient Distributed Training of Sparse and Quantized Models,” preprint *arXiv:2403.09904*, 2024.
>
> [D5] Meinhardt et al. “Sparse-ProxSkip: Accelerated Sparse-to-Sparse Training in Federated Learning,” preprint *arXiv:2405.20623*, 2024.
>
> [D6] Douillard et al. "DiLoCo: Distributed Low-Communication Training of Language Models," *arXiv:2311.08105*, 2023.

---

### Official Review · Reviewer_Hhir · 2025-03-14

**Overall Recommendation:** 4

**Summary:**

This paper proposes FedSMU, a federated learning algorithm that improves communication efficiency and generalization. It symbolizes model updates (using sign-based compression) to reduce communication overhead and mitigate data heterogeneity. Inspired by the Lion optimizer, FedSMU splits local updates and global execution, improving generalization. The performance of FedSMU is validated through both theoretical analysis and empirical experiments.

**Claims And Evidence:**

Most claims in the paper are well-supported by theoretical analysis and empirical experiments.

**Essential References Not Discussed:**

The paper covers most essential references, but it would be better to include Adaptive Federated Learning in the Related Works section for better context.

**Experimental Designs Or Analyses:**

The experimental design is mostly sound, with comparisons across multiple FL baselines. However, a potential limitation is that the experiments primarily focus on small- to medium-scale models (LeNet, ResNet18), leaving uncertainty about FedSMU’s performance on larger architectures.

**Methods And Evaluation Criteria:**

The proposed method is appropriate for the federated learning setting.

**Other Comments Or Suggestions:**

1. In Line 164, there is a typo: "ruducing" should be "reducing".
2. Some notations are not explained in the paper, e.g., $\beta_1$ and $\beta_2$ are momentum coefficients, and $\gamma_2$ is the weight decay factor.

**Other Strengths And Weaknesses:**

The generalization benefits of FedSMU lack theoretical justification.

**Questions For Authors:**

FedSMU splits the standard Lion optimizer into local updates and global execution. How does each component impact FedSMU’s performance?

**Relation To Broader Scientific Literature:**

FedSMU combines communication efficiency and generalization improvements in a unified FL framework, which previous works have addressed separately.

**Theoretical Claims:**

I did not verify the correctness of the proofs for the theoretical claims. However, Assumption 4.3 is not a weak assumption and may not always hold in practice.

---

> ### Author Rebuttal · Authors · 2025-04-01
>
> We would like to thank the reviewer for the comments.
> > Impact of component of FedSMU
>
> As described in Appendix F.9, we had implemented two variants, Fed-LocalLion and Fed-GlobalLion, to evaluate the isolated impact of Lion on the client and server sides. To further assess the necessity of key components in FedSMU, we conduct here a systematic ablation study by removing specific elements and comparing each pruned variant against the full algorithm. In particular, we examine the effect of excluding the following component: 1) server-side weight decay regularization ($\gamma_2$), 2) client-side gradient sliding average ($\beta_1$), 3) client-side gradient symbolization.
>
> The experimental results in Table C1 (https://anonymous.4open.science/r/A-7823/F.pdf) demonstrate that the client-side sliding average plays the most crucial role in achieving a stable and effective training. Additionally, the model update symbolization mechanism itself proves to be more effective than using the full-precision updates. This validates our motivation of proposing FedSMU that symbolization balances contributions of the heterogeneous clients by suppressing some extreme update magnitudes, which thus enhances aggregation stability and leads to better generalization.
>
> Moreover, we observe that directly applying the Lion optimizer on either the client or server side yields suboptimal performance, which indicates that our FedSMU effectively harnesses the benefits of Lion, enhancing the generalization while compressing the communication load.
> > Assumption 4.3
>
> We acknowledge that this is a slightly stronger assumption, ensuring that both the compressed targets and momentum terms (the moving averages of gradients) in our theoretical analysis are bounded. Moreover, this assumption has been adopted in other federated optimization and sign-based compression studies [C1, C2]. Specifically, in [C1], the authors demonstrate the convergence of distributed SIGNSGD with momentum under their Assumption 4, which is also the bounded gradient assumption.
>
> [C1] Tao Sun, et al. ``Momentum ensures convergence of SIGNSGD under weaker assumptions,'' ICML 2023.
>
> [C2] Sashank Reddi, et al. ``Adaptive federated optimization,'' ICLR 2020.
> > Evaluations on larger models
>
> We appreciate this suggestion. Following it, we conduct additional experiments using a larger model (Vision Transformer Small (ViT-S) [C3]) on CIFAR-100, with a Dirichlet distribution of 0.25 to simulate non-IID settings.
>
> Due to time constraints, we focus on comparison with the four strong baselines: FedAvg, SCAFFOLD, FedEF-TopK, and FedLion, which demonstrated higher accuracies in our original Table 2.
>
> As shown in Table A3 (https://anonymous.4open.science/r/A-7823/F.pdf), FedSMU consistently achieves a superior performance with the larger-scale model, further confirming its effectiveness and scalability in more FL scenarios.
>
> [C3] Dosovitskiy, et al. ``An image is worth 16x16 words: Transformers for image recognition at scale,’’ *arXiv:2010.11929* (2020).
> > Adaptive FL as related work
>
> We thank the reviewer for this valuable suggestion, and will incorporate a discussion of Adaptive FL in the Related Work section, as follows.
>
> “Several adaptive algorithms [C4, C5] dynamically adjust global learning rates based on the divergence between local and global models, thereby enhancing generalization performance in federated settings.”
>
> [C4] Reddi, Sashank, et al. "Adaptive federated optimization," ICLR 2020.
>
> [C5] Tong, et al. "Effective federated adaptive gradient methods with non-iid decentralized data," *arXiv:2009.06557* (2020).
> > Theoretical justification of generalization
>
> We acknowledge that the current manuscript does not include a theoretical generalization analysis, as our primary focus is on the design of a communication-efficient optimization strategy, for which we provided a convergence analysis. That is, under the general non-convex settings, FedSMU achieves a convergence rate of $\mathcal{O}(\frac{1}{\sqrt{T}})$, where $T$ is the total number of communication rounds. This theoretical result matches with the convergence rates of existing FL algorithms.
>
> Though a theoretical generalization analysis is not included, FedSMU’s generalization ability is thoroughly validated through extensive experiments. Key designs contributing to this include: 1) the symbolization of local updates, which normalizes client contributions and enhances Magnitude Uniformity (MU), helping alleviate data heterogeneity; and 2) the split-Lion optimizer, which decouples updates into local and global components, combining stability and efficiency to further improve generalization.
> > Typos
>
> Thank you for your correction, and we will correct it to "reducing".
> > Notations
>
> Thank you for your suggestion. We will incorporate the missing symbol definitions into Table 1. Specifically, we will clarify that $\beta_1$ and $\beta_2$ denote the momentum coefficients, while $\gamma_2$ represents the weight decay factor.

---

### Official Review · Reviewer_pLQH · 2025-03-14

**Overall Recommendation:** 2

**Summary:**

This paper proposes a new federated learning algorithm, FedSMU, designed to reduce communication costs and mitigate data heterogeneity. The key idea is to transmit only the sign of local updates for each parameter. Both theoretical analysis and experimental results are provided.

**Claims And Evidence:**

Yes.

**Essential References Not Discussed:**

No.

**Experimental Designs Or Analyses:**

No.

**Methods And Evaluation Criteria:**

Yes.

**Other Comments Or Suggestions:**

NA

**Other Strengths And Weaknesses:**

The motivation and core design of the proposed algorithm are clearly articulated. However, it is unclear what trade-offs are made to achieve communication savings.

**Questions For Authors:**

1. A key concern is that quantizing local updates to one bit may cause the averaged gradient direction to deviate from the steepest descent, potentially increasing the number of communication rounds.

2. The paper claims that FedSMU reduces communication costs by transmitting only one bit per local update for each parameter. However, if this leads to more communication rounds, additional overhead—such as more frequent broadcasts—may offset the savings. In Figure 2, does the reported communication cost include both upload and download bits, or only the upload?

**Relation To Broader Scientific Literature:**

This work may contribute to more efficient communication in federated learning.

**Theoretical Claims:**

No.

---

> ### Author Rebuttal · Authors · 2025-04-01
>
> We would like to thank the reviewer for the comments. In the following, we have provided our detailed responses to these comments.
>
> > Comment 1: The motivation and core design of the proposed algorithm are clearly articulated. However, it is unclear what trade-offs are made to achieve communication savings.
>
> **Response:**
>
> We thank the reviewer for the insightful comment. In our FedSMU, the primary trade-off made to achieve communication efficiency is the loss of precision in the model updates incurred by the 1-bit symbolization, which introduces the quantization noise. This may lead to a slight increase in the number of communication rounds required to reach a target accuracy, as shown in Figure 3 in the Appendix.
>
> However, we would like to emphasize that each communication round in FedSMU incurs a significantly lower communication cost compared to other methods. As a result, even with more communication rounds, the overall communication overhead remains substantially lower, as will be further discussed in our response to Question 2. We will also clarify this trade-off in the revised manuscript.
>
> > Question 1: A key concern is that quantizing local updates to one bit may cause the averaged gradient direction to deviate from the steepest descent, potentially increasing the number of communication rounds.
>
> **Response:**
>
> We appreciate the reviewer’s concern, and acknowledge that quantizing local updates to 1-bit may introduce deviation from the averaged gradient direction, which in turn may result in a larger number of communication rounds needed for convergence, as shown in Figure 3 in the Appendix.
>
> However, we would like to emphasize that our FedSMU significantly reduces the communication cost per communication round, which originates from its highly compressed 1-bit update representation. As a result, even if more communication rounds are needed for convergence, the total communication cost remains substantially lower than that of existing methods, as will be further discussed in our response to Question 2.
>
> > Question 2: The paper claims that FedSMU reduces communication costs by transmitting only one bit per local update for each parameter. However, if this leads to more communication rounds, additional overhead—such as more frequent broadcasts—may offset the savings. In Figure 2, does the reported communication cost include both upload and download bits, or only the upload?
>
> **Response:**
>
> We would like to thank the reviewer for pointing out this issue, which can be addressed as follows.
>
> First, we would like to clarify that the communication cost reported in Figure 2 includes only the uplink (client-to-server) transmission for all the comparison algorithms, which will be explicitly stated in the revised manuscript for more clarity.
>
> Second, the total communication cost per round can be generally estimated by using the following expression, as presented in Literature [B1]:
>
> $Total Communication = Uplink Communication +\alpha ·Downlink Communication, \alpha \in [0,1] $.
>
> In practice, due to the factors such as system asymmetry, caching constraints and protocol limitations, the uplink speed is often significantly lower than the downlink speed, as discussed in Literature [B2]. Consequently, many communication-efficient FL studies [B3, B4] focus merely on minimizing the uplink cost alone.
>
> To have a more comprehensive evaluation of our FedSMU, we followed the setting in [B2] and set $\alpha = 0.1$, and depicted Figure B1 (https://anonymous.4open.science/r/A-7823/F.pdf) to compare the total communication cost including both the upload and download bits. It shows that FedSMU remains communication-efficient even when accounting for the downlink overhead, with a significantly lower total cost compared to the other baselines at comparable accuracy levels.
>
> [B1] Condat, Laurent, Ivan Agarský, and Peter Richtárik. "Provably doubly accelerated federated learning: The first theoretically successful combination of local training and communication compression," *arXiv preprint arXiv:2210.13277* (2022).
>
> [B2] Condat, Laurent, et al. "Tamuna: Doubly accelerated federated learning with local training, compression, and partial participation," *International Workshop on Federated Learning in the Age of Foundation Models in Conjunction with NeurIPS,* 2023.
>
> [B3] Li, Xiaoyun, and Ping Li. "Analysis of error feedback in federated non-convex optimization with biased compression: Fast convergence and partial participation," *International Conference on Machine Learning.* PMLR, 2023.
>
> [B4] Richtárik, Peter, Igor Sokolov, and Ilyas Fatkhullin. "EF21: A new, simpler, theoretically better, and practically faster error feedback," *Advances in Neural Information Processing Systems* 34 (2021): 4384-4396.

---

### Official Review · Reviewer_ZtNk · 2025-03-15

**Overall Recommendation:** 3

**Summary:**

In this paper, the authors propose the FedSMU algorithm to address both communication cost and data heterogeneity challenges in federated learning, through sign-based model compression.

## update after rebuttal

I continue to favor acceptance and will leave my rating unchanged.

**Claims And Evidence:**

1. The motivation is unclear, particularly the deep rationale connecting the three major challenges. The paper integrates multiple techniques, including model update symbolization/sign operation/1-bit compression, sliding average, MU Index, and model compensation. However, each technique addresses a different challenge, and there is no clear connection between them.
2. The paper attempts to tackle multiple issues simultaneously, making its main contributions unclear. I strongly recommend that the authors refine the focus of the paper and clearly highlight its core contribution.
3. Regarding communication efficiency, few-shot or even one-shot FL methods [1][2] have been proposed. The authors should discuss these works in relation to their approach.
4. Client sampling in FL is another important issue [3][4], but it is not adequately addressed in the paper. From my understanding, partial client participant is one of the biased client sampling. However, the authors only consider smaller sampling rate. A discussion on how $\textbf{biased}$ client selection impacts the proposed method would be beneficial.
5. Several low-bit model quantization methods [5][6] have been proposed. Can sign-based 1-bit quantization be effectively applied to larger models? Is the quantization error within an acceptable range when using LLMs?
6. As shown in Figure 2, FedSMU exhibits large fluctuations in the early stages, which appears to contradict the findings in Figure 1(c). Can the authors elaborate on this discrepancy?
7. The evaluated datasets and models are not representative of real-world applications. To fully assess the proposed method, I recommend evaluating it on larger models such as ViT and RoBERTa and larger datasets such as DomainNet and GLUE.
8. The writing needs improvement to meet the standards of a prestigious conference like ICML. The sentence flow is disjointed (e.g., Line 23-24 on the left, Line 25 on the right). Some sentences are overly long (e.g., Line 19-27 on the left). Additionally, there are ambiguous terms, such as “values” in Line 44 on the right—how do smaller values reduce data transmission costs? Do you mean precision? Furthermore, multiple keywords with similar meanings (e.g., symbolic/sign/1-bit compression) should be consolidated for clarity. Sometimes the past tense is used as well.
9. I feel a big gap between the contribution summary and the paragraph from Line 70-90.

[1] Zhang, J., Karimireddy, S. P., Veit, A., Kim, S., Reddi, S., Kumar, S., & Sra, S. (2020). Why are adaptive methods good for attention models?. Advances in Neural Information Processing Systems, 33, 15383-15393.
[2] Ahn, K., Cheng, X., Song, M., Yun, C., Jadbabaie, A., & Sra, S. (2023). Linear attention is (maybe) all you need (to understand transformer optimization). arXiv preprint arXiv:2310.01082.
[3] Chen, W., Horvath, S., & Richtarik, P. (2020). Optimal client sampling for federated learning. arXiv preprint arXiv:2010.13723.
[4] Cho, Y. J., Wang, J., & Joshi, G. (2020). Client selection in federated learning: Convergence analysis and power-of-choice selection strategies. arXiv preprint arXiv:2010.01243.
[5] Ma, S., Wang, H., Ma, L., Wang, L., Wang, W., Huang, S., ... & Wei, F. (2024). The era of 1-bit llms: All large language models are in 1.58 bits. arXiv preprint arXiv:2402.17764, 1.
[6] Malekar, J., Elbtity, M. E., & Zand, R. (2024). Matmul or No Matmul in the Era of 1-bit LLMs. arXiv preprint arXiv:2408.11939.

**Essential References Not Discussed:**

See Claims And Evidence.

**Experimental Designs Or Analyses:**

See Claims And Evidence.

**Methods And Evaluation Criteria:**

See Claims And Evidence.

**Other Comments Or Suggestions:**

See Claims And Evidence.

**Other Strengths And Weaknesses:**

See Claims And Evidence.

**Questions For Authors:**

See Claims And Evidence.

**Relation To Broader Scientific Literature:**

no

**Theoretical Claims:**

The paper assumes that the stochastic gradient is an unbiased estimator of the full gradient and that its variance is bounded. However, these assumptions do not align well with recent findings on minibatch gradient distributions in transformer-based models (as I mentioned before that I am concerned the method is not working well with LLMs). Specifically, prior research ([7-8]) has shown that minibatch gradients in attention-based models follow a heavy-tailed distribution rather than a Gaussian-like assumption with bounded variance. This discrepancy raises concerns about the applicability of the theoretical results, as heavy-tailed gradient noise can significantly impact convergence behavior. A discussion on how the proposed method handles such distributions would strengthen the paper.

[7] Zhang, J., Karimireddy, S. P., Veit, A., Kim, S., Reddi, S., Kumar, S., & Sra, S. (2020). Why are adaptive methods good for attention models?. Advances in Neural Information Processing Systems, 33, 15383-15393.
[8] Ahn, K., Cheng, X., Song, M., Yun, C., Jadbabaie, A., & Sra, S. (2023). Linear attention is (maybe) all you need (to understand transformer optimization). arXiv preprint arXiv:2310.01082.

---

> ### Author Rebuttal · Authors · 2025-04-01
>
> We’d like to thank the reviewer for the comments.
> > Motivation & Contribution
>
> FL involves intertwined challenges where improving one may worsens another. For example, communication-efficient methods may reduce generalization, methods addressing heterogeneity often increase communication, and those balancing both may assume full client participation.
>
> We aim to design an algorithm addressing both challenges while remaining effective under partial participation. To this end, we introduce the Magnitude Uniformity (MU) index, inspired by Jain’s fairness, to quantify consistency of client contributions. We observe that higher heterogeneity reduces MU, but degrades generalization. To mitigate this, we propose model update symbolization, which normalizes update magnitudes to enhance MU, thus implicitly reducing heterogeneity and communication. To address potential accuracy loss from compression, we integrate a sliding average mechanism (inspired by Lion) for improved stability.
>
> Unlike existing methods that trade off one aspect for another, our core contribution is to jointly improve communication efficiency and generalization, while supporting partial client participation.
> > Few-shot FL
>
> One-Shot FL typically trains a global model with a single communication round by ensembling and distilling client models using public data [A1, A2]. While FedSMU takes a multi-round paradigm, it shares similar goals and can potentially incorporate One-Shot strategy.
>
> To our best knowledge, [1, 2] are methodologically different: [1] proposes ACClip for handling heavy-tailed gradients in Transformer, and [2] studies linear Transformers for regression. We show that ACClip is compatible with FedSMU, with provenance improved by FedSMU-ACClip in Table A1 (https://anonymous.4open.science/r/A-7823/F.pdf).
>
> [A1] M. Hasan, et al. "Calibrated one round federated learning with Bayesian inference in the predictive space," AAAI 2024.
>
> [A2] N. Guha, et al. "One-shot federated learning," *arXiv:1902.11175* 2019.
> > Client sampling
>
> We assumed a uniformly random participation (unbiased sampling), allowing to isolate and study effects of data heterogeneity and communication.
>
> Though client sampling is not our main focus, FedSMU is still compatible with biased sampling. We incorporate it with loss-based client selection [3], and show in Table A2 that FedSMU with biased sampling improves performance.
> > Larger models and larger datasets, e.g., DomainNet and GLUE
>
> We conducted additional experiments on CIFAR-100 with ViT-Small model. Four strong baselines are compared: FedAvg, SCAFFOLD, FedEF-TopK and FedLion. Table A3 shows that FedSMU consistently outperforms them on this larger model, providing a strong evidence that our sign-based 1-bit quantization approach generalizes well to more complex models.
>
> Due to time constraints, we have not yet evaluated FedSMU with LLMs on RoBERTa or GLUE, but consider this a promising direction for our future work. We will also include a discussion of centralized low-bit quantization methods [5, 6] in Related Work, to better position our contribution within the FL context.
> > Fluctuations in early stages
>
> We clarify this apparent discrepancy from the following perspectives.
>
> 1)Figs. 1(c) and 2 present different metrics. Fig. 1(c) shows the MU index over communication rounds, while Fig. 2 plots the test accuracy against cumulative communication cost. As such, the x-axes differ in both the scale and meaning, and are not directly comparable.
>
> 2)In Fig. 1(c), FedSMU maintains a stable MU from the start due to symbolization, while FedAvg gradually improves. In Fig. 2, the early accuracy fluctuations in FedSMU are due to the initial quantization noise when the model is still far from the convergence. These fluctuations would diminish as training progresses and when symbolic updates begin to stabilize the training process.
> > Theoretical assumption
>
> We acknowledge that recent studies suggest that mini-batch gradients in attention-based models may follow heavy-tailed distributions, thus challenging the standard assumptions in FL, such as the bounded variance.
>
> In this paper, our theoretical analysis is based on the widely adopted assumptions in centralized and federated learning, particularly for CNNs and LSTMs, which follows the assumption in [A3, A4]. Besides, our evaluation on larger models empirically shows that FedSMU still performs well on ViT-S, which might indicate a practical robustness to the non-Gaussian gradients.
>
> Finally, we totally agree that better aligning the theory with behaviors of Transformer-based models is an important direction, and will briefly discuss this in the revised theoretical section.
>
> [A3] X. Li,  et al. "Analysis of error feedback in federated non-convex optimization with biased compression: Fast convergence and partial participation," ICML 2023.
>
> [A4] X. Huang, et al. "Stochastic controlled averaging for federated learning with communication compression," *arXiv:2308.08165* 2023.

---

> > ### Comment · Reviewer_ZtNk · 2025-04-04
> >
> > Thanks for the additional experiments and clarifications. Most of my concerns have been addressed, and I will increase my score. I strongly recommend that the authors discuss the quantization error in more detail in their revision, especially in the context of LLMs.

---

> > > ### Author Response · Authors · 2025-04-08
> > >
> > > Dear Reviewer ZtNk,
> > >
> > > We would like to express our gratitude once again for the time and effort you have dedicated in reviewing our paper, as well as in the rebuttal and discussion phases. Your insightful comments are invaluable for enhancing our work. In the final version of the manuscript and supplementary material, we will incorporate these additional experiments and related works, as well as expand our discussion on the quantization error, particularly in the context of LLMs.
> > >
> > > Specifically, prior quantization methods for LLMs [5, 6] primarily focus on compressing the weights and activations, where the quantization errors may significantly impact the MatMul operations (e.g., in the attention heads). In contrast, our work targets at the gradient quantization in FL, where the use of 1-bit symbolization emphasizes the direction of updates over their precisions. Our preliminary results when using the ViT-Small model in Table A3 (https://anonymous.4open.science/r/A-7823/F.pdf) indicate that the induced quantization error has a minimal impact on the performance of a transformer model.
> > >
> > > We believe that incorporating our gradient quantization scheme with the established low-bit techniques for weights and activations may offer a compelling path towards the fully quantized LLMs in the federated environments. Exploring this incorporation is inspiringly a next step, and will be considered as a promising direction of our future works.

---

### Decision · Program_Chairs · 2025-05-01

**Decision:**

Accept (poster)

**Comment:**

The reviewers are in agreement that this paper makes a valuable contribution and should be accepted. We encourage the authors to carefully address the reviewers’ comments when preparing the camera-ready revision.